# Environmental and societal costs of maize production decrease by addressing the uncertainty in nitrogen rate recommendations

Francisco Palmero [1] ✉, Eric A. Davidson [2], Kaiyu Guan [3,4], Alison J. Eagle [5], Hannah E. Birgé[6], P. V. Vara Prasad [7], Trevor J. Hefley [8], Jeffrey R. Schussler [9] & Ignacio A. Ciampitti [1] ✉

Excessive crop nitrogen (N) fertilization has negative environmental and social consequences. Using maize grain yield response to nitrogen field trials, we consider the uncertainty surrounding N rate recommendations to demonstrate that fertilizer N rates can be reduced by 12–16% in the US Corn Belt, with negligible risk of maize yield losses. This reduction in N fertilizer applications decrease $N_2O–N$ emissions by 10% and N leaching by 13%, leading to a social benefit of 230–$530 M, due to enhanced air and water quality. Additional N reductions could benefit ecosystems and human health. However, the high risk of yield loss associated with additional N reductions makes this practice unacceptable for farmers. This emphasizes the need for incentive programs that consider the responsibilities and limitations of all actors along the food supply chain.

Maize (*Zea mays* L.) is inefficient at utilizing nitrogen (N), meaning N fertilizer applied to croplands typically drives water pollution and increases nitrous oxide emissions while failing to fully optimize the targeted crop productivity[1]. Addressing this challenge is therefore a pressing priority for securing more sustainable agricultural and environmental systems. A thorough assessment of N application rate uncertainty dynamics, in particular, could provide actionable knowledge to meaningfully improve efficiency and mitigate its adverse effects, yet most cropland N application prescriptions fail to do so.

The United States (US) is a global leader in maize production, not only as the largest contributor to the global supply but also as a hub for agricultural innovation. Given this role, how the US manages N fertilizer–critical for both yield and ecological integrity–has global scale implications for the wellbeing of people and nature. The US produces approximately 30% of the global maize crop[2], with over 75% of that output coming from the Corn Belt[3], a region encompassing parts of Illinois, Indiana, Iowa, Michigan, Minnesota, Missouri, North Dakota, Ohio, Nebraska, and Wisconsin (Fig. 1C).

N fertilizer is a critical input for maize production worldwide. Maize grain yield has steadily increased in the US Corn Belt over the last half of the 20th century[4,5], largely due to a combination of plant breeding and changes in crop management practices such as N fertilization and plant density, among other relevant technological advances[6,7]. The invention of the Haber–Bosch process, which converts atmospheric N into reactive forms of N[8], facilitates the use of thousands of tons of manufactured N fertilizer annually for maize production in the US Corn Belt[9]. Therefore, maize productivity is

[1]Department of Agronomy, Purdue University, West Lafayette, Indiana, IN, USA. [2]Appalachian Laboratory, University of Maryland Center for Environmental Science, Frostburg, MD, USA. [3]National Center for Supercomputing Center, University of Illinois at Urbana Champaign, Urbana, IL, USA. [4]Department of Natural Resources and Environmental Sciences, University of Illinois at Urbana Champaign, Urbana, IL, USA. [5]Environmental Defense Fund, Raleigh, NC, USA. [6]The Nature Conservancy, Arlington, VA, USA. [7]Department of Agronomy, Kansas State University, Manhattan, KS, USA. [8]Department of Statistics, Kansas State University, Manhattan, KS, USA. [9]Schussler Ag Research Solutions, Marion, IA, USA. ✉e-mail: fpalmero@purdue.edu; iciampit@purdue.edu

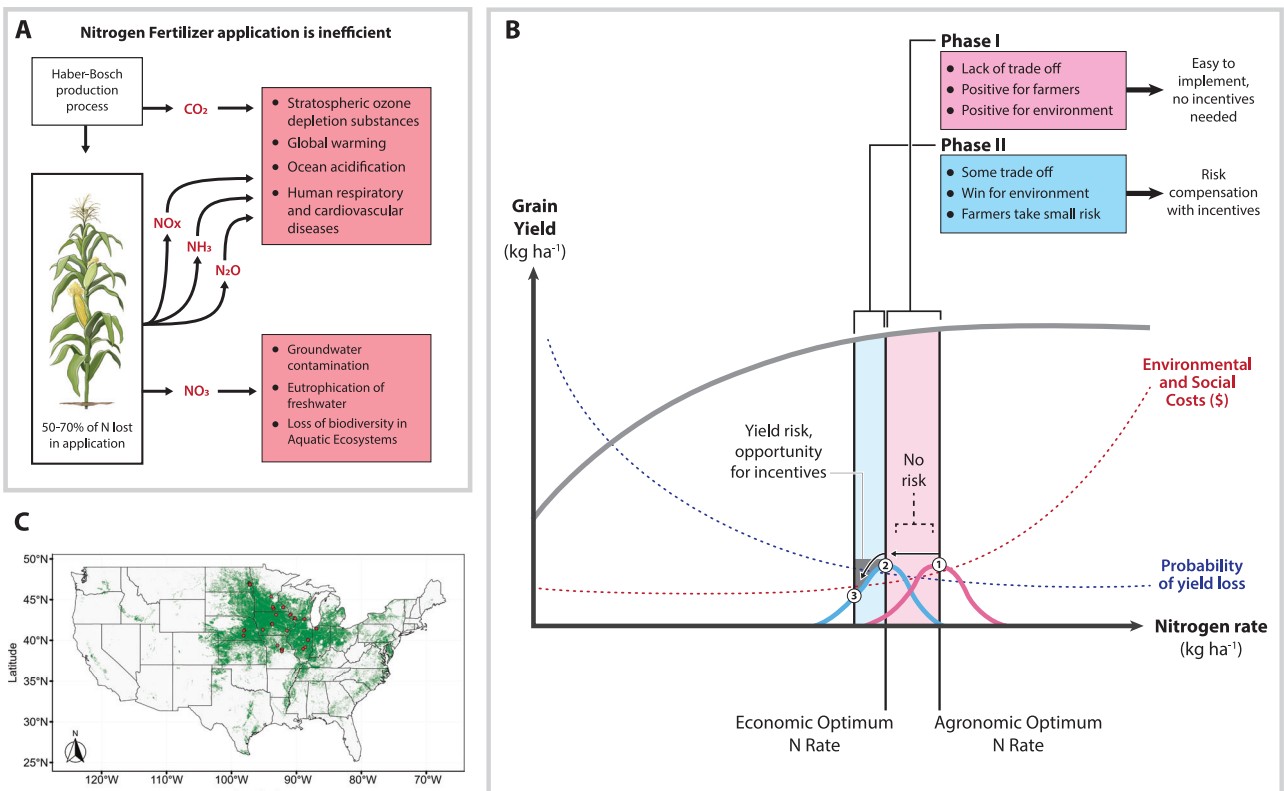

**Fig. 1 | Schematic representation of excessive nitrogen (N) application, statistical framework analysis, and site locations. A** Schematic representation of environmental and human consequences associated with gas emissions and nitrate leaching linked with N fertilization in maize. **B** Schematic representation of the methodological approach implemented in this study. **C** Geographic location of the 49 site-year combinations of all field trials derived from the database (red points). The green color indicates the area cultivated with maize and the gray color the non-maize cultivated area[94].

strongly linked to N fertilization[10,11], which, along with other fertilized crops, makes N fertilization key to nourishing the growing human population[12]. N fertilizer has unwanted impacts for society as well, as N not taken up by crops is vulnerable to oxidation and release into the atmosphere as potent greenhouse gas nitrous oxide ($N_2O$)[13] and as a pollutant in downstream municipal and ecological water supplies[14].

Indeed, upwards of 50 to 70 percent of the total N applied as fertilizer is lost through leaching and runoff (largely as nitrate, $NO_3^-$) and/or in gaseous forms as $N_2O$, $NO_x$, $N_2$, and/or $NH_3$[15–17] (Fig. 1A). The loss of N via $NO_3^-$ to ground and surface waters drives groundwater contamination, eutrophication of aquatic ecosystems and the attendant loss of biodiversity and ecological functioning[18,19], and contamination of municipal water supplies. When municipal water supplies are not properly cleaned of excess nitrate, communities experience a host of health challenges, including increased risk of cancer[20], birth defects[21], and overall mortality[20].

Soil bacteria naturally fix dinitrogen gas into ammonium in the soil, where it is vulnerable to oxidation into $NO_3^-$ and other forms of N. Under low oxygen conditions, $NO_3^-$ is reduced by different soil bacteria into $N_2O$ and other N species. $N_2O$ from the soil leaks into the atmosphere, where it acts as a potent greenhouse gas (GHG) and stratospheric ozone-depleting substance[22,23]. Soil bacteria also release N oxide ($NO_x$) and ammonia ($NH_3$) gases during these processes which contribute to tropospheric ozone, formation of fine particulate matter ($PM_{2.5}$), and $CH_4$ oxidation[24,25]. While this cycling of N occurs in all terrestrial systems, the presence of additional N in the soil from fertilizer amendments accelerates the production of these unwanted gases. Additionally, the production of N fertilizer via the Haber–Bosch process is energetically intensive and typically relies on fossil fuel

energy[26]. These undesirable environmental impacts of fertilizer production and use negatively affect human health, crop productivity, food security, and economic prosperity[27–29].

Despite the unwanted social and environmental impacts of excess soil N, most crops are in fact N limited due to a mismatch in the timing, location, and type of N vis-a-vis plant needs[30]. Nitrogen limitations are well documented to decrease crop productivity, and with it, farmer and rural economic wellbeing[31]. Clarifying the relationship between maize grain yield and N fertilization is essential for improving N management with the goal of minimizing its adverse environmental and social consequences while optimizing yield and farmer economic return[32,33]. This relationship, along with maize and N fertilizer prices, determines the optimum N input needed to maximize yield (agronomic optimum N rate; AONR) or farmers' profit (economic optimum N rate; EONR). However, fluctuations in soil moisture, soil N status, and crop growth dynamics drive considerable uncertainty on the determination of the AONR and EONR[34–37]. This uncertainty in the recommended N application rates can be represented by probability distributions (Fig. 1B). Left unaddressed, this uncertainty can lead to constantly recurring over or under application of N to maize crops, further exacerbating the problem of N inefficiency and its associated ills. In this study, we modeled the relationship between maize grain yield and N fertilizer rate using a quadratic-plateau model, illustrated in Fig. 1B.

Most current fertilization programs recommend a single N rate that maximizes farmers' profit applied either to the entire field or to sub-regions within it[38]. However, farmers recognize the uncertainty inherent in those recommendations and typically apply more N than recommended to hedge against yield loss[39,40]. This practice is an insurance for farmers to make sure they can obtain excellent yields

whenever weather and other conditions are favorable, with application N rates resulting in higher levels than those recommended in the US Corn Belt[41,42]. The uncertainty of N rate recommendations is usually overlooked in studies that evaluate the production, environmental, and social impacts of changes in N management[11,43–49]. Ignoring uncertainties might lead to inaccurate conclusions about yield losses when the applied N rate is reduced[11]. Therefore, uncertainties for defining the optimal N input should be accounted for when studying the societal and environmental impacts of this practice. A more holistic and transparent handling of the uncertainty around yield-related metrics of N efficiency resulting from this approach offers actionable insight into making marginal improvements to the N efficiency problem in agriculture.

To demonstrate the need for a more explicit accounting of uncertainty in crop N-efficiency dynamics, we analyzed different cases of N fertilizer reduction in maize across eight US Corn Belt states (Illinois, Indiana, Iowa, Minnesota, Missouri, Nebraska, North Dakota, and Wisconsin) while accounting for the uncertainty around optimal N rates and economically quantifying the savings in environmental costs to society under the most feasible case of N rate reduction (Fig. 1B). Based on the fact that farmers apply N beyond the recommendations, we explored two phases of N reduction.

The probability of losing yield, the expected yield loss, and the environmental and social costs were evaluated (Fig. 1B). In the first phase (Phase I) the reduction goes from the expected value of the AONR (E[AONR]) to the expected value of the EONR (E[EONR]) implying economic gains for farmers and positive environmental impacts (Fig. 1B). This reduction is still conservative since a high proportion of farmers apply more N than the recommended N rate, which is usually above the E[AONR][50–52]. In the second phase (Phase II), four cases of N reduction were evaluated based on the quantiles of the probability distribution of the EONR. Thus, in this phase, the N cutback goes from the E[EONR] to some quantile of the EONR distribution so that farmers take the smallest reasonable risk of economic losses (Fig. 1B). Therefore, our analysis covers more than profit-maximizing scenarios, spanning from N rates maximizing grain yield to the probability distribution of optimal N rates that maximize profits. Although the risk of yield and economic losses in this Phase II is very small, farmers have the perception of being "caught short"[39]. Therefore, in this phase additional incentives will be needed to deal with the way farmers traditionally think[50–52]. By assuming that farmers apply E[AONR] in Phase I, our analysis remains both realistic and conservative relative to current N application rates[50–52].

## Results

### Fertilizer N rates: point and uncertainty estimations

The quadratic plateau model fitted to the data captured different shapes on the maize grain yield response to N fertilization in the 49 sites analyzed in the database (Fig. S1). Out of these sites, 17 were removed because of poor convergence of the AONR and/or EONR, and/or yield plateau was achieved at a N rate (E[AONR]) greater than the highest rate implemented in the studies (Table S2).

To facilitate the interpretation of the results, we first describe the calculations for point and uncertainty estimations considering a particular site and then summarize the general results across all sites. As an illustrative example, we selected site T18, corresponding to a field experiment conducted in Lewis, Iowa, during the 2015 season. For this site, the estimated E[AONR]–the mean of the posterior probability distribution of the AONR–was 149 kg ha$^{-1}$ (Fig. S2). The 0.025 and 0.975 quantiles of this posterior distribution, which define the 95% credible interval and thus the uncertainty of the estimate, were 114 and 191 kg ha$^{-1}$, respectively. The difference between these two quantiles determines the uncertainty of the AONR estimation, which resulted to be 77 kg ha$^{-1}$, representing ~52% of the E[AONR] (Fig. S2). The same procedure was applied to all sites to estimate both the point values

(expected values) and the associated uncertainties of the AONR and EONR.

Across all sites, the E[AONR] ranged from 108 to 291 kg ha$^{-1}$ with an average of 198 kg ha$^{-1}$ (Fig. S2). The difference between the 0.025 and 0.975 quantiles (95% credible interval) determines the uncertainty of the AONR estimations. The uncertainty of the AONR had an average of 97 kg ha$^{-1}$ with a minimum of 54 kg ha$^{-1}$ and a maximum of 163 kg ha$^{-1}$ (Fig. S2). Proportional to the E[AONR], the uncertainties on the estimations of the N rates that maximized maize grain yield were between 27 and 87%. The expected value of the N rate that maximizes farmers' profit, i.e., E[EONR], had a mean of 180 kg ha$^{-1}$ with a minimum of 101 kg ha$^{-1}$ and a maximum of 261 kg ha$^{-1}$ (Fig. S3). The EONR uncertainty was, on average, 46% (range from 26 to 74%), with a minimum of 49 kg ha$^{-1}$ and a maximum of 138 kg ha$^{-1}$. These numbers highlighted a substantial uncertainty for the N rate that maximizes grain yield (AONR) and farmers' profit (EONR) in maize production across different sites of the US Corn Belt region (Figs. S2 and S3).

### Feasibility of Nitrogen fertilization reductions

To facilitate interpretation, we first describe the calculations of the expected grain yield loss and its associated probability in Phase I, where the N rate was reduced from E[AONR] to E[EONR]. For each iteration of the Markov Chain Monte Carlo algorithm (Bayesian model fitting), E[AONR] and E[EONR] were estimated, and their corresponding yields were predicted. Because these predictions were functions of random variables (model parameters and optimal N rates), they generated two probability distributions—one for yield at E[AONR] and another for yield at E[EONR]. The difference between these two distributions yielded a third distribution representing yield change, from which we derived the expected yield loss (the mean), its 95% credible interval, and the probability of yield loss. To express yield losses as proportions, the expected yield loss was divided by the expected yield at E[AONR]. Further methodological details are provided in the "Conceptual framework" subsection of the Methods.

As an illustrative example, we revisit site T18, a field experiment conducted in Lewis, Iowa, during the 2015 season. For this site, the expected yield loss was −24 kg ha$^{-1}$, with a 95% credible interval from −159 to 272 kg ha$^{-1}$ (Fig. S4). The probability of yield loss was 59%, i.e., P(yield at E[AONR] > yield at E[EONR]) = 0.59, and by complement, P(yield at E[EONR] > yield at E[AONR]) = 0.41. The expected yield at E[AONR] was 11,711 kg ha$^{-1}$, corresponding to a relative yield loss of 0.20% in Phase I (Fig. S4). The same procedure was applied to all sites in both Phases I and II, regarding the changes in optimal N rates according to the studied Phase.

Across all sites, In Phase I, the N reduction was evaluated as the change from the E[AONR] to E[EONR]. In this phase, on average, there was 62% chance of losing between 11 to 68 kg ha$^{-1}$ of maize grain yield across the evaluated sites (Fig. S4). This reduction was observed with an expected grain yield ranging from 8507 to 16,537 kg ha$^{-1}$ at E[AONR]. Therefore, in Phase I, there was 53 to 0.71% probability of losing between 0.08 and 0.49% of maize grain yield (Fig. S4). Thus, there were two implications in this phase. First, the N rate was reduced from the expected value of the N rate that maximizes maize grain yield to the expected value of the N rate that maximizes farmers' profit, meaning that farmers would earn money due to reducing N application. Second, there was a probability smaller than 71% of losing less than 0.49% of yield (economically not relevant), i.e., yield losses in this Phase I were disregarded.

In Phase II, the expected yield losses and their corresponding probabilities increased from case 1 to case 4 (quantiles 0.4 and 0.1 of the EONR distribution, respectively) (Fig. S5). The expected yield loss among the different cases substantially changed depending on the type of maize grain yield response to N fertilization (sites). However, changes in probability among the different scenarios were similar regardless of the site (Fig. S5).

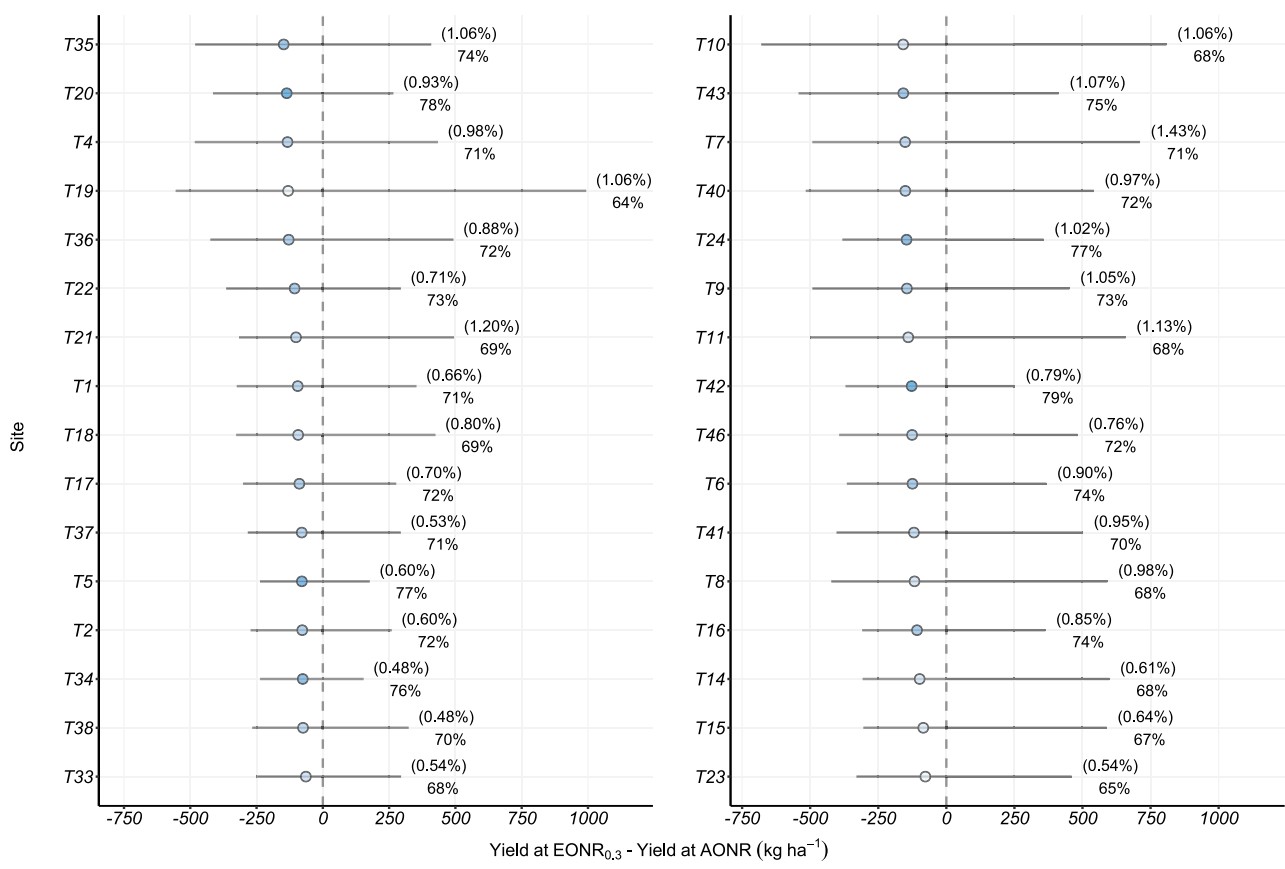

**Fig. 2 | Yield loss and its associated probability when reducing the nitrogen (N) rate from the expected value of the agronomic optimum nitrogen rate (AONR) probability distribution (E[AONR]) to the 0.3 quantile of the economic optimum N rate probability distribution (EONR$_{0.3}$).** This reduction in N fertilizer input corresponds to the total N fertilization reductions considering Phases I and II. Circles indicate the expected yield loss; the horizontal bars indicate the 95% credible interval of the yield loss estimation. On the right of each horizontal bar, the upper-level values, between parentheses, indicate the proportion of yield loss with respect to yield at E[AONR], and the lower-level values on the right and circle colors (darker referring to higher probabilities) indicate the probability associated with yield loss estimations. The labels in the $y$-axis indicate the number of the trial in the original dataset, which are referred to as site-by-year combinations in this study. In each site, the total number of observations was 32 ($n = 32$) with four repetitions per treatment. Source data are provided as a Source Data file.

The combination of the probability of losing yield with the expected yield loss can be thought of as the risk taken by farmers when reducing the N rate. In all of the sites, case 2 (quantile 0.3 of the EONR probability distribution) was a low risk or the safest zone since it implied, on average, 74% chances of expected yield loss between 39 to 132 kg ha$^{-1}$. In case 2, the estimated expected value of yield at E[EONR] had a maximum of 16,708 kg ha$^{-1}$ and a minimum of 8438 kg ha$^{-1}$. Therefore, the previously mentioned yield losses (ranging between 39 to 132 kg ha$^{-1}$) represented grain yield reductions between 0.45 and 1.06% relative to yield at E[EONR]. This means that, under case 2, there was a 70 to 79% probability of losing between 0.45% and 1.06% of grain yield (Fig. S5). These numbers highlighted case 2 as a feasible case for reducing N rates in the US Corn Belt, independently of the maize grain yield response to N fertilization.

Finally, when considering the two phases of N reduction altogether, the N reduction was from E[AONR] to the quantile 0.3 of the EONR distribution (case 2). In the total N cutback from fertilization, there was 71% probability of losing between 64 and 158 kg ha$^{-1}$ relative to the yield at E[AONR] (Fig. 2). In other words, considering the total N reduction suggested in this study, there was 64 to 79% probability of losing between 0.48 and 1.43% of grain yield (Fig. 2). The probability of yield loss was similar across the evaluated phases and when combining Phase I and II, but there was a little increase in the proportion (and the amount) of expected yield loss. The results shown here depict the absence of (Phase I) or low risk (Phase II) of economic losses and should be considered when dealing with farmers' perception of being caught short of N when fertilizing maize with this nutrient.

## Nitrogen rate reduction: representative scenarios

Considering the uncertainty of the AONR and EONR (Figs. S2 and S3), three sites were selected as representative scenarios of high, medium, and low uncertainty of N rate recommendations for evaluating the environmental and social impact of N fertilization reductions suggested in Phases I and II. The sites "T10" (Troth, Missouri, in 2014 season), "T46" (Kyes, Nebraska, in 2016 season), and "T5" (Loam, Indiana, in 2014 season), were selected and named "Scenario A" (high uncertainty), "Scenario B" (medium uncertainty), and "Scenario C" (low uncertainty), respectively.

The maize grain yield response curves to N fertilization for the three selected scenarios are presented in Fig. 3A. The crop rotations over the previous four years consisted of continuous soybean in Scenario A and maize–soybean rotation in Scenarios B and C. Scenario A represented high uncertainty in the N rate recommendations (163 and 138 kg ha$^{-1}$ for the AONR and EONR, respectively), Scenario B an

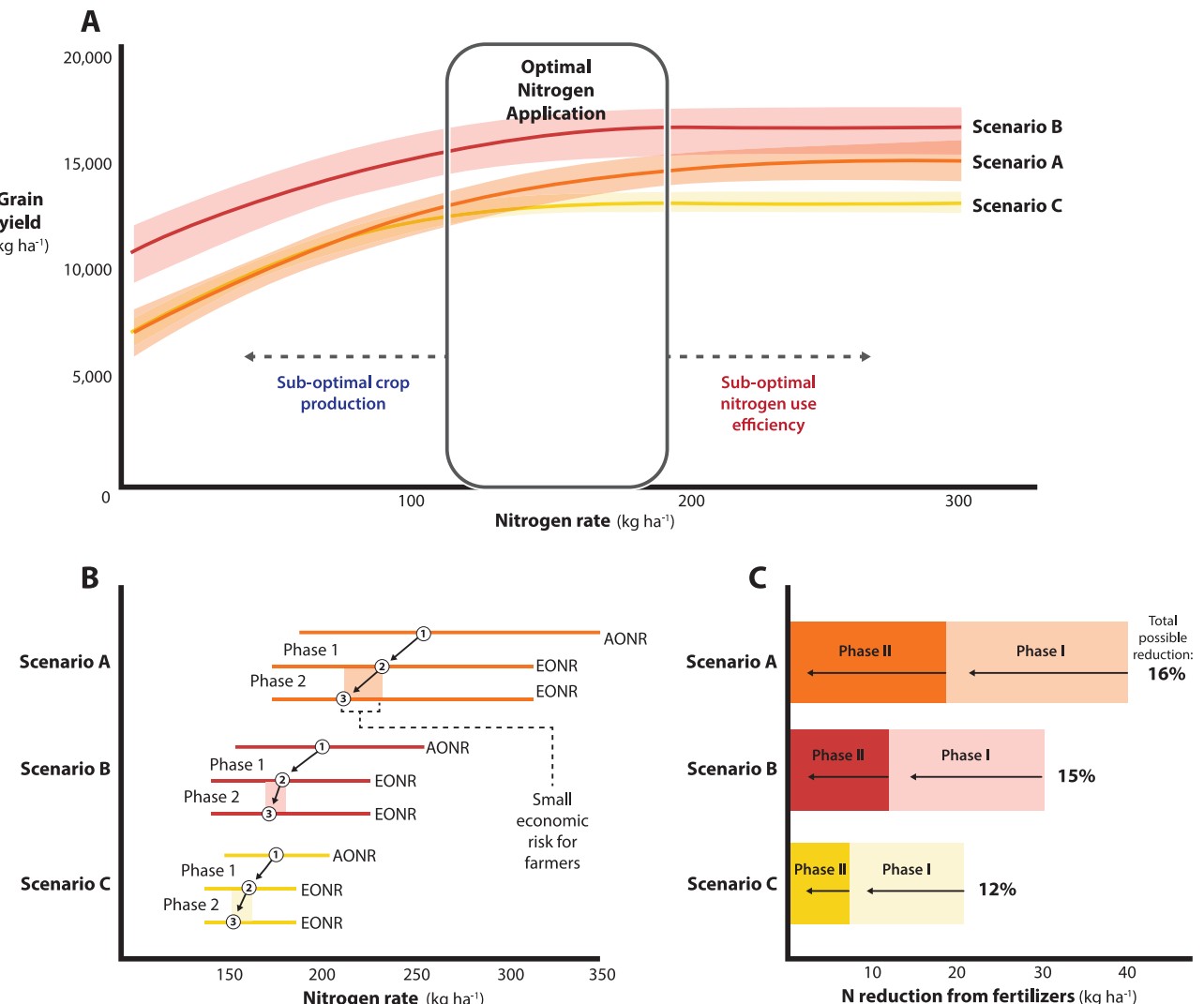

**Fig. 3 | Nitrogen (N) reduction analyses in the three selected scenarios.**
**A** Relationship between maize grain yield ($y$) and nitrogen rate ($x$) in each of the selected scenarios. The solid lines represent the expected grain yield at different fertilizer N rates. The shadow areas indicate the 95% credible interval of the posterior predictive distributions. **B** Summary of the agronomic optimum N rate (AONR) and economic optimum N rate (EONR) for each of the selected scenarios. Circles with number one indicate the expected value of the AONR distribution, i.e., E[AONR]. Circles with number two indicate the expected value of the EONR distribution, i.e., E[EONR]. Circles with number three indicate the quantile 0.3 of the EONR distribution. The horizontal bars indicate the 95% credible interval of the AONR and EONR distributions. **C** Nitrogen reduction from fertilizers when reducing the N rate from the E[AONR] to the E[EONR] (Phase I), and from E[EONR] to the 0.3 quantile of the EONR (Phase II). Numbers on the right indicate the total N reduction of fertilizer application rates relative to the E[AONR]. Source data are provided as a Source Data file for panels (**B**, **C**).

average uncertainty (103 kg ha$^{-1}$ in the AONR and 85 kg ha$^{-1}$ in the EONR), and Scenario C low uncertainty (56 and 49 kg ha$^{-1}$ for the AONR and EONR, respectively) (Fig. 3B and Figs. S2 and S3). The range of expected values for the AONR was between 174 kg ha$^{-1}$ (Scenario C) and 254 kg ha$^{-1}$ (Scenario A), with a middle value of 205 kg ha$^{-1}$ (Scenario B). Furthermore, E[EONR] ranged from 161 kg ha$^{-1}$ to 231 kg ha$^{-1}$, with a middle value of 182 kg ha$^{-1}$ for Scenarios C, A, and B, respectively (Figs. 3A and 2B).

In Phase I, the N reduction from E[AONR] to E[EONR] was 19 kg ha$^{-1}$ in Scenario B, with a maximum of 22 kg ha$^{-1}$ and a minimum of 14 kg ha$^{-1}$ in Scenarios A and C, respectively (Fig. 3B, C). Proportional to the E[AONR], these reductions represented 9% (Scenario A) and 8% (Scenarios B and C). In the second phase, the N reduction from E[EONR] to the 0.3 quantile of the EONR had a maximum of 19 kg ha$^{-1}$, middle of 12 kg ha$^{-1}$, and a minimum of 7 kg ha$^{-1}$ observed in Scenarios A, B, and C (Fig. 3B, C). Proportional to the E[EONR], these N cutbacks constituted 7% (Scenarios A and B), and 4% (Scenario C).

Considering the reductions in Phases I and II, a total N reduction between 21 and 41 kg ha$^{-1}$ (with middle of 31 kg ha$^{-1}$) could be achieved (Fig. 3C). That is, it was possible to reduce the N rate between 12 and 16% (Fig. 3C) without (Phase I) or with minor risk (Phase II) of yield losses in the US Corn Belt when uncertainties are properly accounted for defining N input for maize.

As the AONR and EONR uncertainties increased (Fig. 3B), the N reductions in Phases I and II also increased (Fig. 3C). Furthermore, as the expected value of the optimum N rate increased the uncertainty increased (Fig. S6). These results indicate that it should be expected that sites with higher optimum N rates have higher uncertainty and a greater potential to reduce the N fertilization rates with no (Phase I) or small risk (Phase II) of yield losses.

## Environmental and social gains
The N fertilization reduction was translated into environmental and social benefits. In Phase I, the $NO_3^- - N$ leaching was reduced by an

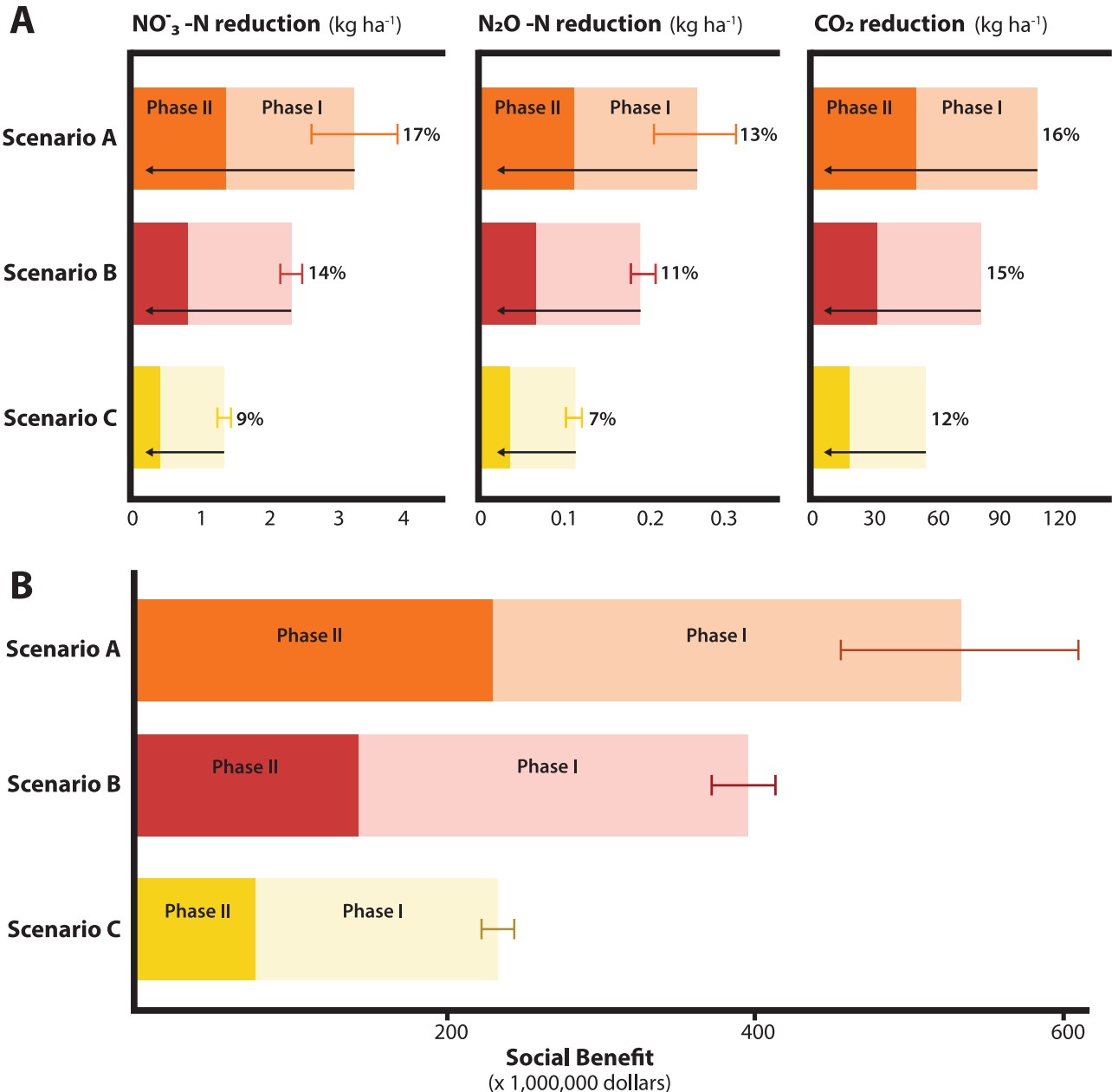

**Fig. 4 | Environmental and social benefits associated with nitrogen (N) fertilizer reductions.** In **A**, the height of the bars indicate the expected $NO_3^- - N$, $N_2O$-N, and $CO_2$ reductions when reducing the fertilizer N input from the expected value of the agronomic optimum N rate to the expected value of the economic optimum N rate (Phase I), and from the expected value of the economic optimum N rate to the 0.3 quantile of the economic optimum N rate (Phase II). Numbers on the right indicate the total emission or N leaching reduction relative to emissions or leaching at the expected value of the agronomic optimum N rate. **B** The height of the bars indicate the expected social benefit associated with N reductions in Phases I and II, and the total. The whiskers indicate the standard error of the estimated total social benefit. In **A**, **B**, the total number of observations was 32 ($n = 32$) with four repetitions per treatment for each scenario. Source data are provided as a Source Data file.

average of 1.42 kg ha⁻¹ (Fig. 4A). The maximum $NO_3^- - N$ leaching reduction was 1.84 kg ha⁻¹ (Scenario A) and the minimum 0.90 kg ha⁻¹ (Scenario C). This represented a cutback of $NO_3^- - N$ leaching between 10 to 6% with an average of 8% (Fig. 4A). The N fertilization cutbacks found in Phase II reduced the $NO_3^- - N$ leaching by 0.83 kg ha⁻¹ (0.41 kg ha⁻¹–1.38 kg ha⁻¹) (Fig. 4A), representing an additional 5% (3–7%) reduction with respect to the emissions at the E[EONR]. In the case of $N_2O$–N, the reductions in Scenario A, B, C were 0.15 kg ha⁻¹ (7%), 0.13 kg ha⁻¹ (7%), and 0.08 kg ha⁻¹ (5%), respectively in Phase I (Fig. 4A). While in Phase II, the average $N_2O$–N emission reduction was 0.08 kg ha⁻¹, with a minimum of 0.04 kg ha⁻¹ and a maximum of 0.12 kg ha⁻¹ (Fig. 4A). Lastly, the $CO_2$ emissions associated with

fertilizer manufacturing and transport were reduced between 37 and 58 kg ha⁻¹ in Phase I, and between 18 and 50 kg ha⁻¹ in Phase II (Fig. 4A). In summary, overall reductions were up to 17% for the $NO_3^- - N$ leaching, 13%, for $N_2O$–N emissions, and 16% for the $CO_2$ emissions associated with both Phases with minor impact on crop productivity and economic returns for farmers.

For economically quantifying the social benefits of reducing the N rate in maize, the equivalent $CO_2$ and N leaching reductions were scaled to the maize planted area across the eight US states. Considering a price of 18.54 \$/kg N leaching and 50 \$/t $CO_2$ (see Methods section), the N fertilization cutbacks resulted in a social benefit between \$156 M and \$298 M (average of \$234 M) in Phase I and

between \$74 M and \$231 M (average of \$148 M) in Phase II (Fig. 4B). Therefore, properly accounting for the uncertainty in the N rate recommendations to farmers brought a social benefit of up to \$529 M with the absence of (Phase I) or low risk (Phase II) of economic losses in the US Corn Belt. This social benefit was likely underestimated since the social benefit of reducing $NO_x$ and $NH_3$ emissions was not considered.

## Discussion

In this study, we analyzed phases of N fertilizer reduction for the most relevant producing region of US maize, while accounting for the uncertainty of the recommended N rates to farmers. Reducing N fertilizer application between 12 and 16% in the US Corn-Belt region had a concomitant average reduction of 10% in $N_2O$–N emissions and 13% of $NO_3^- $ – N leaching. Furthermore, this environmental gain was translated into a societal benefit, on average, close to \$400 M. The approach implemented in this study considered the risk of yield losses to evaluate the balance of maintaining acceptable grain yields for farmers while minimizing the undesirable environmental consequences of N fertilization. This approach can be applied provided that the optimal N rates are estimated with reasonable accuracy and precision. Otherwise, overly optimistic or inflated N reduction estimates may arise, potentially biasing the conclusions of this study. Here, we lay the foundation for establishing best management practices (BMP) for reducing N rates in maize not only in the US but other regions with N overapplication around the globe. These analyses should be the first step in setting the basis for any environmental, social, and policy studies analyzing the risk:benefit ratio of N fertilization in agriculture.

### Uncertainty on the fertilizer N rate recommendations

A substantial uncertainty of the optimal N rates was found in this study, regardless of the maize grain yield response to N applications. The estimated uncertainty was between 27 and 87% for the AONR and between 26 and 74% for the EONR. In terms of kilos per hectare, average uncertainties of 97 and 81 kg ha$^{-1}$ were found on the AONR and EONR estimations, respectively. The uncertainty on the optimal N rates for maize were previously reported to be greater than 29% in the US Corn Belt[34,53]. In this area, different studies found an EONR uncertainty ranging between 25–73 kg ha$^{-1}$[37], 16–204 kg ha$^{-1}$[53], 8–261 kg ha$^{-1}$[34], and 131–208 kg ha$^{-1}$[35], and an AONR uncertainty between 140–234 kg ha$^{-1}$[35]. Another source of uncertainty in estimating optimal N rates arises from our limited ability to identify the true optimum—an uncertainty that cannot be quantified numerically. This limitation is evident at sites exhibiting linear grain yield responses to N fertilizer (-12% of the analyzed sites in our study), where increasing N consistently enhances maize productivity. Although an optimal N rate can still be estimated under such conditions, these estimates are typically upward-biased and highly uncertain. In practice, when grain yield responds linearly to N, farmers, crop advisors, and extensionists recognize that additional N will increase yield (and profits), but the precise rate required remains unclear. Such situations might often lead to excessive N applications.

When the grain yield response to N is accurately represented by a model with a decreasing slope (e.g., quadratic-plateau functions), reductions in the N rate lead to an increase in the secant slope between the yield at zero N application and the yield at a given N rate. Therefore, under these conditions, lowering N rates inherently improves fertilizer use efficiency. In contrast, when the yield response to N is better described by a model with a constant slope (e.g., a linear response), reducing the N rate does not alter the secant slope between the yield at zero N application and the yield at a given N rate. In such cases, lowering N does not enhance fertilizer use efficiency and may substantially increase the risk of yield loss. Consequently, these conditions introduce additional uncertainty that cannot be fully quantified and may lead to N overapplications.

Recent studies determined the environmental optimum N rate, which targets the mitigation of the societal cost of N fertilization due to N externalities. These studies reported a difference of 28[49] and 70 kg ha$^{-1}$[111] between the EONR and the environmental optimum N rate, and of 3 kg ha$^{-1}$ between the AONR and EONR[11] in the US Corn-Belt. These values are within the uncertainty estimation of the AONR and EONR reported in this and previous studies. Therefore, the variability in factors such as soil N budgets measurement, weather forecast, crop N demand, the price of N fertilizers and maize, and environmental costs suggests the existence of an *optimal N fertilization zone* where the AONR, EONR, and the environmental optimum N rate lie. Usually, farmers decide to maximize "good years" reducing the chances of converting those into less good years[40]. However, the *optimal N fertilization zone* represents a range of "optimal" N rates allowing farmers to evaluate the probability of losing yield (or money) and the expected yield (or money) loss, i.e., the risk, associated with making one or another decision. Lower N rates within the *optimal N fertilization zone* will lead to lower environmental costs associated with increments in farmers' profit over time because of the good and not good years ratio in a time series[54,55].

### Feasibility of N reductions: economic, environmental, and social aspects

In this article, we evaluated not only the expected yield loss associated with N fertilization reduction but also the probability associated with those losses. This allowed us to comprehensively study the risk of reducing N fertilizer applications in the US Corn-Belt. A recent study reported that reducing N fertilization by 29 kg ha$^{-1}$ caused a reduction of 3% in maize grain yield in the US Corn-Belt[11]. In other cases, cutting the N rate 33 kg ha$^{-1}$ had no impact on maize grain yield[56] or a reduction of 45 kg ha$^{-1}$ led to a yield loss of 2%[57]. In this article, we demonstrated comparable results of the impact of similar N rate reductions on maize grain yield in the US Corn Belt. Most importantly, we quantified the probability associated with these yield losses, showing a negligible risk of diminishing maize grain yield when reducing up to 16% of the N fertilizer inputs.

Since the probability of losing yield is not zero, experiencing yield losses is still a possible event. Then, in the worst case, the US maize production would decrease 0.4 million tons, representing 0.1% of the total US maize production in the 2024 season[2]. Considering economic losses because of yield reduction and economic savings due to reducing N application, this maize production would represent, on average, an economic loss of 7 \$ ha$^{-1}$ for each farmer. However, the societal benefit associated with reducing N losses to the atmosphere and groundwater would be, on average, 16 \$ ha$^{-1}$, offsetting any potential economic yield losses. Net social benefits above 1000 \$ ha$^{-1}$ have been previously reported in the US Corn-Belt[44,49]. Funds saved from sites with positive net social benefits could be redirected toward environmental protection incentives, which could be used to compensate farmers in the unlikely event of yield losses.

When analyzing the unlikely situation of yield losses in Phase II, scenarios of high and medium uncertainty (Scenarios A and B, respectively) had an economic loss of 3 and 4 \$ ha$^{-1}$ and a social benefit of 10 and 6 \$ ha$^{-1}$, respectively. Then the net social benefit was 7 \$ ha$^{-1}$ (Scenario A) and 2 \$ ha$^{-1}$ (Scenario B). In Scenario C (low uncertainty), the net social benefit was −0.5 \$ ha$^{-1}$. However, these net social benefits were underestimated since potential reductions in $NO_x$–N and $NH_3$–N emissions were not accounted for[43]. Regions where the net social benefit is positive are identified as "hot spots", these are areas with high potential for N losses mitigation[44]. We showed that higher E[AONR] or E[EONR] were associated with higher uncertainty, and higher uncertainties were related to higher N rate reductions. Therefore, even in the worst case within the low-risk case of N reductions (Phase II), sites with high uncertainty in the AONR or EONR would still

be areas with high potential for mitigating N losses. Identifying and understanding the spatial variability of the uncertainty in the optimum N rates recommendation would be key to find and define more precisely where the hot spot areas are located.

Although the above situation has a low chance of being observed, farmers still face a small risk of economic losses when making N reductions within Phase II. However, incentives to environmental protection can be developed to encourage farmers to reduce their N fertilizer rates[58]. Funds saved from sites with positive net social benefits could be used to compensate farmers for yield losses elsewhere. Therefore, identifying zones with high and low uncertainty on the optimal N rate estimations will be crucial for designing incentives for achieving N rate fertilization reductions in the US Corn-Belt. Furthermore, in this study, we considered that farmers applying more N than it is recommended applied an N rate corresponding to the E[AONR], while there are cases where farmers apply more than the E[AONR][50,52]. Thus, the identification of hot spots for N loss mitigation should be based on current N application and the uncertainty of the optimal N rate recommendations.

### Further reductions of N environmental fallout

The use of N fertilizers with reactive N has increased since 1970s in crop production[19]. When N fertilizers are excessively applied, undesirable environmental consequences occur[18,19,22,23]. Accordingly, N reduction assessments in crop production have received considerable attention via governing leader resolutions[59,60]. Soil organic carbon is an important reservoir of carbon in croplands, reducing the amount of $CO_2$, a greenhouse gas, in the atmosphere[61,62]. Furthermore, soil organic carbon is related to crop productivity[63,64]. Since crop productivity is unlikely to be affected by the N rate reductions proposed in this study, changes in soil organic carbon are unlikely to occur. Therefore, we propose N fertilization changes that lower greenhouse gas emissions without compromising other mitigation strategies in agriculture. In the first phase of N fertilizer input reduction, we found that reducing the N rate ~8% decreased 6% $N_2O-N$ emissions without economic losses for farmers. The shift from E[AONR] to E[EONR] inherently provides environmental benefits, as it involves reducing the N rate relative to that required to maximize grain yield. In Phase II, a reduction of 6% in the N rate reduced 4% $N_2O-N$ emissions with a negligible risk of yield and economic losses for farmers. Since the N fertilization rates were not above the E[AONR], it was expected that in both phases, the reductions of the N rate were proportionally higher than that of the $N_2O-N$[43] due to the non-linear relationship between $N_2O-N$ emissions and N balance[65].

Kim et al[44]. reported that reductions of 10% on the N rate application led to reducing 2 to 17% the $N_2O-N$ emissions and 6 to 13% the $NO_3^- - N$ leaching with yield losses lower than 1.3% in the US Corn-Belt without providing a significant yield decline. Another study observed that reducing the N rate between 29 and 79 kg ha$^{-1}$ decreased N losses ($N_2O-N$ emissions + $NO_3^- - N$ leaching) by 3 to 12 kg ha$^{-1}$, with an associated yield loss of 3% – 14%, but with no measurement specifying how likely the yield losses were[11]. In our study, we found that reducing the N application rate by 12 to 16% resulted in 7 to 13% reductions in $N_2O-N$ emissions and 9% to 17% in $NO_3^- - N$ leaching with no or negligible risk (low probability and low expected value) of yield losses when accounting for the uncertainty of the N rate recommendations to farmers.

Other alternatives (beyond reducing the N rate) to decrease $N_2O-N$ emission and $NO_3^- - N$ leaching in maize production were not explored in this article. The N management practices considering spatial variability[66] and the right rate, right source, right placement, and right timing ("4R") can also be implemented to mitigate N losses without necessarily modifying the amount of N applied to the crop[67,68]. Beyond these changes in crop management practices, there might be opportunities to reduce $N_2O-N$ emissions and $NO_3^- - N$ leaching by

implementing environmental policy tools such as voluntary N credit programs, group incentives, or even N insurance[69,70].

Voluntary N credit programs might consist of using agroecosystem and machine learning models[44] or empirical relationships[71] to monitor reductions in $N_2O$, $NO_x$, and $NH_3$ emissions, and $NO_3^-$ leaching due to N cutbacks in the fertilizer applications. Then, for example, reductions in greenhouse gas emissions ($N_2O$, $NO_x$, and $NH_3$) can be transformed to $CO_2$ equivalent, making 1 ton of $CO_2$ equivalent to 1 credit. Finally, farmers can get economic compensation for reducing the N rate by selling the credits to companies who emit higher amounts of greenhouse gases[72]. The group incentives would imply economic rewards for groups of farmers based on greenhouse gas emissions and water quality, which are determined by the combined actions of all the farmers in a given area or watershed[73]. The N insurance program could consist of the payment of a premium by farmers. In the case that the farmers lose yield due to reducing the N fertilizer application (with respect to a reference N rate), the organization providing warranties would indemnify farmers[74].

### Calling for sciences integration

The environmental consequences of excessive N application in croplands negatively impact human health, crop productivity, food security, and economic prosperity[27–29]. Alternatively, under-applications of N fertilizers decrease crop productivity, affecting human nutrition, soil degradation, and farmers' profit[31]. Furthermore, global crises such as the COVID-19 pandemic and geopolitical conflicts in Eastern Europe have altered the fuel-fertilizer-transportation-food nexus at a global level. Consequently, N fertilizer prices have increased, modifying farmers' practices in regions such as the US Corn Belt, where the proportion of maize planted relative to soybeans has declined. This cascade of events affected commodity prices, risking food security worldwide[75].

These facts demonstrate the role of N management to tackle global food challenges. N management studies designed to identify BMPs to mitigate N losses in agricultural fields should be addressed by interdisciplinary teams considering the entire process (industrial production through crop utilization) involved in the N fertilization practice[76,77]. Potential N reductions beyond those proposed in this study can be attractive from an ecosystem and human health point of view, but unattractive to farmers due to the high risk of yield losses, thus making further N reductions unsustainable. Therefore, to effectively tackle N pollution reductions from croplands, governmental and industry programs should be designed considering the responsibilities and limitations of all the parties along the food chain[70,77].

### Summary and future research directions

Considering the uncertainty of the AONR and EONR allowed us to study the risk of yield losses associated with reducing N fertilizer applications in maize. This analysis balanced the achievement of acceptable grain yields for farmers while minimizing the undesirable environmental consequences of N fertilization. We showed that accounting for the uncertainty on the N rate recommendations might allow farmers to apply less N with no or small risk of yield losses. These would lead to a reduction in surface and groundwater contamination, in the loss of biodiversity and ecological functioning, in the contamination of municipal water supplies, and in greenhouse gas emissions. This would be translated into healthier environments and a lower risk of human diseases, resulting in an economic and social benefit.

For a given region, current N recommendation programs provide farmers with a single value of EONR or AONR, or with a range of values empirically defined according to profit considerations. These N rates might be suitable for some fields, but not for others. Alternative approaches have been implemented where the optimum N rates are defined specifically for different regions within a field. The second

approach represents an improvement in terms of reaching a more efficient use of N. However, both methods are still far too simplistic because both recommend single N rates without considering the uncertainty associated with it.

Almost all the effort in N fertilization studies has been allocated to obtain precise and accurate estimations of a single value of the optimum N rates. Although this is necessary, it is not enough. Rather, a high research priority should be on dissecting the uncertainty of N rate recommendations. The influence of crop rotation on uncertainty estimation—and consequently on N rate reductions and their associated risk of yield loss—merits further investigation in subsequent research. Furthermore, the uncertainty is mainly driven by the interaction among soil N dynamics, weather conditions, and crop growth dynamics. Our capacity to capture different interactions of soil, weather, and crop characteristics is growing with new satellite sensors, sensor-integrated machines, and computational power. The task is still very challenging due to the high complexity of the agroecosystems. Even when two similar soils and crops are considered, the N rate maximizing crop yield or farmers' profit will change depending on what moment of the crop cycle the precipitation or other weather events occur. Nevertheless, merging the concepts of post-season N rates (such as grain yield response to N) with in-season N rate adjustment (for example, based on the N nutrition index) and site-specific N rate recommendations may provoke new measurement and modeling approaches for better understanding N rate uncertainties.

## Methods
### Database description
In this study, we utilized the open dataset from the Performance and Refinement of N Fertilization Tools (PRNT), a public–industry partnership project presented by refs. 78,79.

This database contains 49 site-years combinations of plot research studying the maize grain yield response to N fertilization. The experiments were conducted during the 2014, 2015, and 2016 growing seasons across eight states in the US Corn Belt (Iowa, Illinois, Indiana, Minnesota, Missouri, North Dakota, Nebraska, and Wisconsin) (Fig. 1). Hereafter, the site-year combinations are referred to as "site". The same experimental protocol was carried out at each site. For this study, we filtered the cases in which the N rates were applied at planting (broadcast to the soil surface). A total of 8 N rates from 0 to 315 kg ha$^{-1}$ were applied in increments of 45 kg ha$^{-1}$. A randomized complete block design with four repetitions was implemented in each site. For crop sampling, the maize grain yield (kg ha$^{-1}$) was obtained in each experimental unit, and it was adjusted at 155 g kg$^{-1}$ moisture. Furthermore, the grain N content (kg ha$^{-1}$) was also obtained in each experimental unit by multiplying the grain N concentration and the grain dry matter. At 96% of the sites, a maize–soybean rotation was implemented, while the remaining 4% consisted of continuous maize or soybean.

Soil characteristics and prevailing weather conditions exert a strong influence on maize yield responses to applied nitrogen[80]. To capture this variability, two contrasting sites were selected within each of eight US states for each of the three study years (2014–2016): one site located on a highly productive soil and a second site characterized by comparatively lower soil productivity. Collectively, these locations spanned a substantial portion of the US Corn Belt and encompassed a broad range of soil types and climatic regimes[78]. The selected sites were broadly distributed across the three dominant soil orders in the region—Alfisols, Mollisols, and Entisols.

Mean annual precipitation exhibits considerable spatial variability across the study region, increasing by nearly a factor of two from the northwestern to the southeastern areas. In addition to this spatial gradient, precipitation shows a pronounced seasonal pattern, with ~70–80% of total annual rainfall occurring during the spring and summer months of the maize growing season (April–October)[78]. Mean

annual air temperatures also vary substantially across the region, generally increasing along a north–south gradient. As a consequence of this temperature variability, the length of the growing season differs markedly among sites.

To account for site-specific differences in growing season duration, maize hybrids were selected to ensure appropriate comparative relative maturity ratings and other agronomic traits suited to each location. This selection strategy aimed to maximize yield potential while reducing the risk of frost damage at individual sites. The hybrids used in the study ranged in comparative relative maturity rating from 89 to 115 days.

### Conceptual framework
The grain yield response to N fertilization curves is affected by crop management, soil, and weather factors. Different curves can be obtained across seasons, even within the same location[32]. Therefore, the "optimal" N rate has considerable uncertainty[34–37]. Our uncertainty attributed to estimating the response curve, and consequently on the AONR and EONR, for a specific season in a given site can be represented by the posterior distribution of curves using Bayesian inference. We fitted a quadratic plateau model to the grain yield response to N fertilization via Bayesian inference. Therefore, instead of having a point estimation of the model parameters (with its respective uncertainty on the estimation), we had a probability distribution of the parameters. This allowed us to obtain a probability distribution of the optimum N rates (i.e., AONR and EONR) (Step 1 – Fig. 5). In Phase I, the N reduction was from E[AONR] to E[EONR], that is, from the red vertical line to the blue vertical line, as represented by the horizontal black arrow in Step 1 – Fig. 5. In Phase II, four cases of N reduction were created based on the EONR distribution. Furthermore, the posterior predictive distribution for grain yield was obtained for each case (Step 1 – Fig. 5). These distributions were used to compute the probability of losing yield and the expected yield loss with respect to yield at the E[EONR] in a given curve within a given site (Step 2 – Fig. 5).

These steps enabled us to evaluate the feasibility of reducing the recommended N rates considering the AONR and EONR uncertainties, i.e., regarding that at planting it is not known which is the appropriate N rate for a specific season.

Steps 1 and 2 were repeated in each of the studied sites presented in this article. In Phase II, the most feasible case for N reduction across sites was identified based on the probability of losing yield and the expected yield loss. After this, we selected representative sites based on the amount (kg ha$^{-1}$) and the proportion (%) of N reductions in Phases I and II. In those sites, we translated the N rate reduction into savings in air ($N_2O$–N emissions), and water ($NO_3^-$ – N leaching) pollution (Step 3 – Fig. 5). Finally, the sustainability of the change in the N fertilization practice was estimated by computing the social benefit (Step 4 – Fig. 5).

### Statistical analysis
Bayesian inference was used in this study to perform probabilistic inference while accounting for the uncertainty in the parameter estimation, and to easily obtain the uncertainty of model predictions.

A quadratic plateau model (see its shape in Step 1 – Fig. 5) was fitted to data for the relationship between maize grain yield (response variable; $y$) and N rates (predictor variable; $x$). This model allowed us to analytically obtain the AONR by taking the first derivative of the model, equating it to zero, and solving for $x$. Then

$$AONR = \frac{\beta_1}{-2\beta_2}, \tag{1}$$

where $\beta_1$ is the slope of the model, and $\beta_2$ is the quadratic term. Following a similar procedure, but equating the first derivative of the model to the price ratio (PR), the EONR (Step 1 – Fig. 5) can be

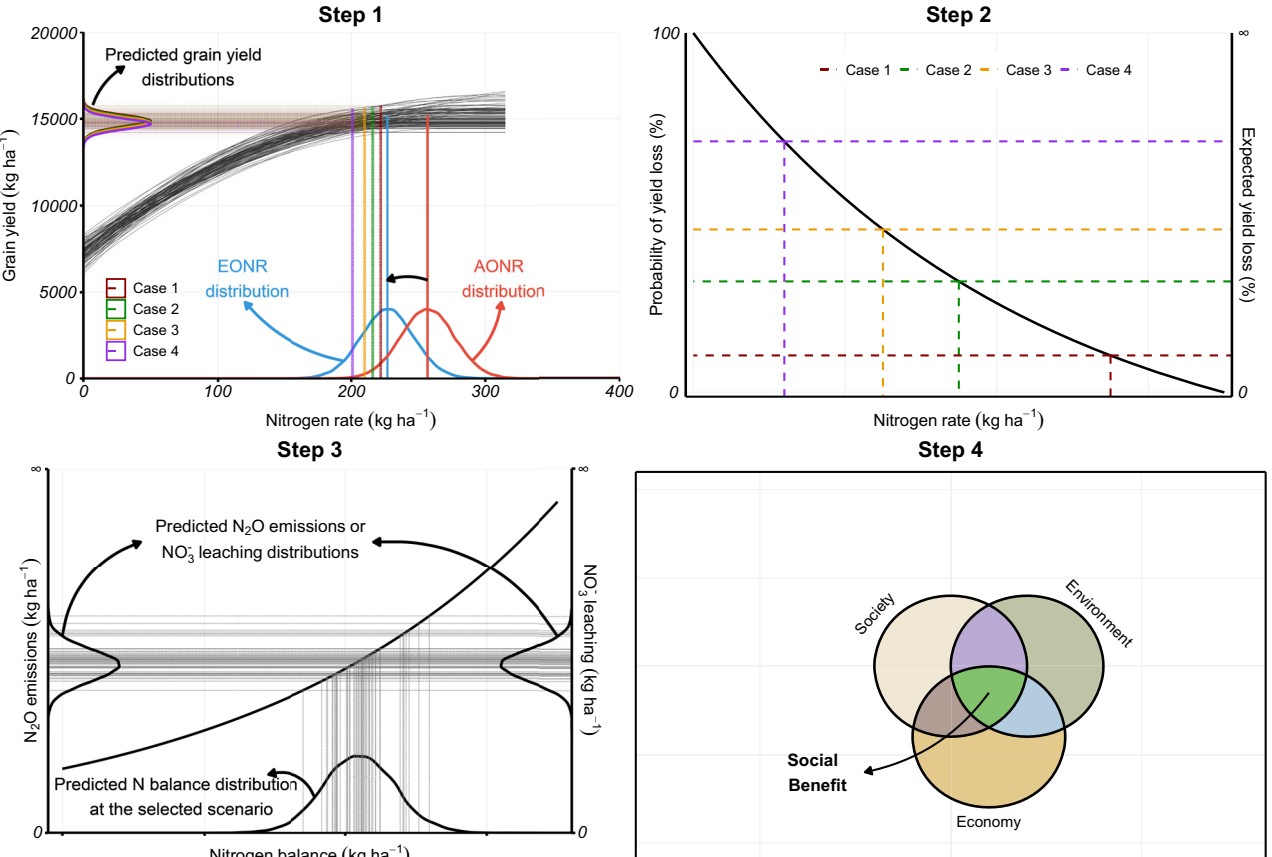

**Fig. 5 | Conceptual framework. (Step 1)** Gray solid lines represent 100 quadratic plateau models fitted to the data of maize grain yield (*y*) versus nitrogen rate (*x*) in a given site using Bayesian inference. In Steps 1 and 2, cases 1, 2, 3, and 4 represent the 0.4, 0.3, 0.2, and 0.1 quantiles of the EONR probability distribution, respectively. **(Step 2)** The probability of yield loss and the expected yield loss with respect to yield at the E[EONR] in a given curve (those presented in Step 1) for each of the four

created cases of nitrogen fertilizer reduction. **(Step 3)** Estimation of nitrous oxide ($N_2O-N$) emissions and nitrate ($NO_3^- - N$) leaching using nitrogen balance for the selected case of nitrogen fertilizer reduction in Step 2 as a predictor variable. **(Step 4)**. Social benefit is computed to estimate the social impact and the sustainability of the change in $N_2O-N$ emissions and $NO_3^- - N$ leaching by changes in the N fertilization practice.

obtained as

$$EONR = \frac{\beta_1 - PR}{-2\beta_2} \quad (2)$$

The *PR* is calculated as $PR = \frac{Nprice(\$/kg)}{maize\,price(\$/kg)}$, i.e., PR represents the kg of maize needed to pay for 1 kg of N of a given fertilizer. To include the uncertainty on the PR, we implemented the approach utilized by ref. 34. Maize grain and fertilizer prices were obtained from the USDA-Economic Research Survey[9,81]. For maize grain, the price was considered as future price in April for the time period 2000–2021. For the fertilizers, the price of anhydrous ammonia (82-0-0) and urea (46-0-0) in April or March was considered for the same time period. The PR was calculated for each year included in the search. Based on PR mean (5.34 kg maize/kg N) and variance (1.87 kg maize2/kg N2), it was assumed that $PR \sim gamma(15.2, 2.8)$. Where 15.2 and 2.8 are the shape and the rate of the distribution, respectively. That is $E[PR] = 15.2/2.8$, and $var(PR) = 15.2/2.8^2$. Therefore, instead of considering the PR as a fixed known quantity, each time the EONR was calculated, the PR was sampled from $gamma(15.2, 2.8)$.

The quadratic plateau model fitted to the grain yield and N rate data in this study was:

$$y_{ij} \sim normal(\mu_{ij}, \sigma_\varepsilon), \quad (3)$$

$$\mu_{ij} = \begin{cases} \beta_0 + \tau_j + \beta_1 x_i - \beta_2 x_i^2, \text{ if } x_i < AONR \\ \beta_0 + \tau_j + \beta_1 AONR - \beta_2 AONR^2, o.w. \end{cases}, \quad (4)$$

$$\beta_0 \sim gamma\left(\alpha_{\beta_0}, \beta_{\beta_0}\right), \quad (5)$$

$$\beta_1 \sim gamma\left(\alpha_{\beta_1}, \beta_{\beta_1}\right), \quad (6)$$

$$\beta_2 \sim gamma\left(\alpha_{\beta_2}, \beta_{\beta_2}\right), \quad (7)$$

$$\tau_j \sim normal(0, \sigma_b), \quad (8)$$

$$\sigma_b \sim uniform(0, 5000), \quad (9)$$

$$\sigma_\varepsilon \sim gamma\left(\alpha_{\sigma_\varepsilon}, \beta_{\sigma_\varepsilon}\right), \quad (10)$$

where $y_{ij}$ is the grain yield (kg ha$^{-1}$) for the i$^{th}$ observation in the jth block, $x_i$ is the ith observation of the N rate (predictor variable), $\mu_{ij}$ is the underlying process, $\beta_0$ (intercept) represents grain yield when no N is applied, $\beta_1$ (slope) is the parameter modeling the linear response of maize grain yield to N rate, $\beta_2$ is the parameter modeling the quadratic effect of the N rate on maize grain yield. Finally, $\tau_j$ is the

random effect of the jth block on $\beta_0$, $\sigma^2_b$ is the variance of $\tau_j$, and $\sigma^2_\varepsilon$ is the variance of maize grain yield. This model was fitted to the data for each of the six sites within the selected states. The Eqs. (5–10) are the prior distribution of the model parameters, where the Greek letters or numerical values within the parentheses are called hyperparameters. The values for the hyperparameters were defined for each state. For each parameter following gamma distribution, the expected value can be obtained as $\alpha/\beta$ and the variance as $\alpha/\beta^2$.

The hyperparameters, except those for $\sigma^2_b$ in Eq. (9), were obtained by fitting a quadratic plateau model to the data utilized in ref. 34 for maize grain yield versus N rate relationship. The model was fitted to Iowa, Illinois, Indiana, Minnesota, Missouri, North Dakota, Nebraska, and Wisconsin separately. The model was also Bayesian, and the priors were set as wide uniform distributions to reduce their influence on the posterior distribution of the model parameters. The same priors were used in each state. Subsequently, the expected value and the variance of each parameter were obtained from their posterior distributions and utilized to build the hyperparameters for the model fitted in the current study. Therefore, the hyperparameters in Eqs. (5–8) and Eq. (10) changed according to the states and those can be found in Table S1. From Eqs. (1–10), it can be seen that the AONR and EONR are functions of the model parameters ($\beta_1$ and $\beta_2$), which are considered as random variables in the Bayesian framework. Therefore, the AONR and EONR are derived quantities and Monte Carlo integration can be used to approximate their probability distributions.

In addition, to estimate the GHG emissions and $NO_3^- - N$ leaching at different N rates, it was necessary to determine the N exported in maize grains at a given N rate (Step 3 – Fig. 5). Therefore, a quadratic or quadratic plateau model was fitted to the relationship between the grain N content (kg ha⁻¹) and N rates in each site. These models can be written as

$$y_{ij} \sim normal\left(\mu_{ij}, \sigma_\varepsilon\right), \tag{11}$$

$$\mu_{ij} = \beta_0 + \tau_j + \beta_1 x_i - \beta_2 x_i^2, \tag{12a}$$

or

$$\mu_{ij} = \begin{cases} \beta_0 + \tau_j + \beta_1 x_i - \beta_2 x_i^2, & if\, x_i < \left(\frac{\beta_1}{2\beta_2}\right) \\ \beta_0 + \tau_j + \beta_1\left(\frac{\beta_1}{2\beta_2}\right) - \beta_2\left(\frac{\beta_1}{2\beta_2}\right)^2, & o.w. \end{cases}, \tag{12b}$$

$$\beta_0 \sim uniform(0, 300), \tag{13}$$

$$\beta_1 \sim uniform(0, 5), \tag{14}$$

$$\beta_2 \sim uniform(0, 2), \tag{15}$$

$$\tau_j \sim normal\left(0, \sigma_b\right), \tag{16}$$

$$\sigma_b \sim uniform(0, 1000), \tag{17}$$

$$\sigma_\varepsilon \sim uniform(0, 1000). \tag{18}$$

In this case $y_{ij}$ is the grain N content (kg ha⁻¹) for the ith observation in the jth block, $x_i$ is the ith observation of the N rate (predictor variable), $\mu_{ij}$ is the underlying process, $\beta_0$ (intercept) represents grain N content without N application, $\beta_1$ (slope) is the parameter modeling the linear effect of the N rate on grain N content, $\beta_2$ is the parameter modeling the quadratic effect of the N rate on grain N content. Lastly,

$\tau_j$ is the random effect of the jth block on $\beta_0$, $\sigma^2_b$ is the variance of $\tau_j$, and $\sigma^2_\varepsilon$ is the variance of grain N content. Equations (13)-(18) are the prior distribution for the model parameters, which were set so that their influence on the posterior distributions was reduced as much as possible.

The plots for the models fitted in Eqs. (3–10) and Eqs. (11–18) are shown in Figs. S1 and S7. The posterior predictive distribution of each model presented in Eqs. (3–10) and Eqs. (11–18) was studied to evaluate model performance (Figs. S8 and S9). The posterior predictive distribution represents the uncertainty in the model parameters and the ability of the model to make accurate predictions of the data[82].

The models were fitted to the data to obtain the posterior distributions of the parameters using a Markov Chain Monte Carlo (MCMC) algorithm. Four chains were run in parallel with a total of 60,000 iterations and a warm-up period of 30,000 iterations. After warm-up, 1 in 10 draws was saved (i.e., thinning = 10) to reduce autocorrelation among draws before making inference. The convergence of the models was explored via trace plots and Gelman−Rubin diagnostic[83]. The models were fitted in R program[84] (version 4.3.3), using the *rjags* package[85]. The code to fit the models is available at ref. 86. All the data implemented in this study is freely available in different repositories (see Data availability).

## Assessing N reduction feasibility

Assuming farmers apply N so that yield is maximized, i.e., AONR[41–43], the feasibility of reducing the N fertilization rate in the US Corn Belt was analyzed by a probabilistic assessment of different cases. The uncertainty on the AONR, the EONR, and on maize and fertilizer prices were considered (Step 2 – Fig. 5).

After fitting the models to the data, the model parameters, AONR, and EONR (derived quantities) convergence were analyzed. Furthermore, the E[AONR] was obtained and considered as the most likely N rate at which yield was maximized in each site. We assumed chain convergence when chains mixed well in trace plots and Gelman−Rubin diagnostic was between 1 and 1.02[83]. Sites showing Gelman−Rubin diagnostic >1.02 were not considered for further analysis. Additionally, if yield plateau was achieved at a N rate (E[AONR]) greater than the highest rate implemented in the studies, the site was not considered as selectable to avoid extrapolating results beyond the range of N rates explored in the studies. These criteria also allowed us to avoid relatively high values of N reduction from the E[AONR] to the E[EONR]. Based on these criteria, 17 out of the 49 sites were regarded as not reliable sites to make inference, therefore they were removed before further analyses (Table S2).

Including sites where the yield plateau was reached beyond the tested N rates required extrapolation to estimate the optimal rates, thereby inflating uncertainty and producing unrealistically large N reductions with minimal yield risk. However, a high proportion of such uncertainty would arise from limitations in experimental design or data quality rather than from inherent variability in weather, soil, or crop demand. Similarly, sites with Gelman−Rubin values above 1.02 typically exhibited no response or a linear yield response, indicating model non-convergence and unreliable estimates of optimal N rates. Therefore, these sites were excluded to provide a more conservative and realistic assessment of potential N rate reductions and their associated risk of yield loss.

In the 32 remaining sites, the N rate reduction from the E[AONR] to the E[EONR] were computed. Moreover, to determine the feasibility of further N reductions, the probability distribution of the EONR was considered to represent its uncertainty. Four cases of N reduction were created based on different quantiles of the EONR distributions (Step 1 – Fig. 5). For cases 1, 2, 3, and 4, the grain yield posterior predictive distribution at quantiles 0.4, 0.3, 0.2, and 0.1 were obtained, respectively. Since the posterior distribution of the EONR was symmetric in all sites, quantile $0.5 \approx$ E[EONR], then an incremental N reduction was

considered from case 1 to case 4. Afterward, the posterior predictive distributions at each quantile were implemented to compute the probability of losing yield at each case with respect to yield at the E[EONR] in a given curve within a given site. The most feasible case for N fertilization reduction was defined based on the probability of losing yield and the expected yield loss (Step 2 – Fig. 5). After this, three sites (out of the 32) were selected as the most representative based on the amount (kg ha$^{-1}$) and the proportion (%) of N reduction from E[AONR] to E[EONR] (Phase I) and from E[EONR] to some quantile of the EONR (Phase II) representing the most feasible case of N reduction. The selected sites were referred to as "scenarios" (Scenario A, B, and C).

### Environmental and social impact

More efficient N utilization creates multiple material benefits for society. In the US Corn Belt, farmers almost universally hedge their risk by over-applying, which means more efficient utilization is almost invariably achieved via N rate reduction in this region—when the suitable model for the grain yield response to N fertilization has a decreasing slope. The N rate reduction was translated into a social benefit under the "safest" case of N rate reduction in the selected scenarios (Step 4 – Fig. 5). For calculating the social benefit (SB), we considered the social cost of N fertilization associated with GHG emissions and water pollution from N loss to ground and surface water. According to ref. 44, the SB was calculated as:

$$SB\left(\$ha^{-1}\right) = \left(GHG_{\Delta N_2O} + GHG_{\Delta fert}\right)P_{GHG} + \Delta N_{leaching}P_{leaching}, \quad (19)$$

where $GHG_{\Delta N_2O}$ (kg ha$^{-1}$) is the reduced N$_2$O–N emission linked to the N rate reduction, $GHG_{\Delta fert}$ (kg ha$^{-1}$) represents the reduction of the CO$_2$ emission associated to fertilizer manufacturing and transport, $P_{GHG}$ is the social cost of GHG emissions ($/t CO$_2$), $\Delta N_{leaching}$ is the reduction in N leaching, and $P_{leaching}$ is the social cost for N leaching ($/kg N). McLellan et al[87] and Eagle et al[65] proposed models to estimate N leaching ($NO_3^- - N$) and N$_2$O–N emissions in agricultural fields based on the relationship between soil N balance with $NO_3^- - N$ leaching and with N$_2$O–N emissions (Step 3 – Fig. 5). It was assumed that $NO_3^- - N$ leaching and N$_2$O–N emissions followed gamma distributions with expected value $\alpha/\beta$ and variance $\alpha/\beta^2$. The models proposed by refs. 87,65 were implemented to model the expected values as

$$E[N_2O - N]\left(kg\,ha^{-1}\right) = e^{(0.339 + 0.0047\,N\,balance)}, \quad (20)$$

$$E[NO_3^- - N]\left(kg\,ha^{-1}\right) = e^{(2.459 + 0.0061\,N\,balance)}, \quad (21)$$

where

$$N\,Balance\left(kg\,ha^{-1}\right) = total\,N\,applied\left(kg\,ha^{-1}\right) \\ - grain\,N\,content\left(kg\,ha^{-1}\right). \quad (22)$$

The total N applied (kg ha$^{-1}$) is the N from mineral fertilizer and other N inputs such as manure, N-fixing cover crops, irrigation water, etc. Since the other N inputs beyond N from mineral fertilizer were not present in the experiments considered in this study, totalNapplied = Nrate. The grain N content (kg ha$^{-1}$) is the N removed in maize grains. In this study, the grain N content was obtained from the posterior predictive distribution of the models for grain N content vs N rate evaluated at the E[AONR] and at the N rate for the selected case in each of the three scenarios (i.e., N rate for the 0.3 quantile of the EONR distribution). Then, the N balance, the $E[NO_3^- - N]$ leaching, and the $E[N_2O - N]$ emissions were estimated for the E[AONR] and for the selected case in each of the three scenarios. These estimations were computed for each value in the posterior predictive distribution of the models for grain N content vs N rate.

We conducted a literature review to retrieve information about N$_2$O–N emissions and $NO_3^- - N$ leaching in the US Corn Belt. The search was conducted through Web of Science®. The 31 of June 2024 was selected as a cut-off date, after which literature searches were no longer conducted. The implemented keywords were: "fertilizer", "nitrogen" OR "N", "agriculture", "nitrous oxide", "emissions", "nitrate", and "leaching". The selection criteria were: (i) the experiment was conducted in the studied states within the United States of America (Iowa, Illinois, Indiana, Minnesota, Missouri, North Dakota, Nebraska, and Wisconsin); (ii) the experiments were performed in field conditions; (iii) only corn-based systems were considered; (iv) the implemented N rates were between 50 kg N ha$^{-1}$ and 300 kg N ha$^{-1}$ because N$_2$O–N emissions and $NO_3^- - N$ leaching were calculated at the E[AONR], E[EONR], and the 0.3 quantile of the EONR; and (v) manure were not included because of uncertainty and variability in nutrient composition. Out of the retrieved articles, a total of 31 studies were considered to summarize N$_2$O–N emissions and $NO_3^- - N$ leaching in the US Corn Belt (Table S3).

The variances for $NO_3^- - N$ leaching and N$_2$O–N emissions in the US Corn Belt were obtained from a literature review (Table S3) and it was assumed fixed over the scenarios. The $E[NO_3^- - N]$, the var($NO_3^- - N$), the $E[N_2O - N]$, and the var($N_2O - N$) were implemented to calculate the parameters of the gamma distribution as $\alpha' = E[.]^2/\text{var}(.)$ and $\beta' = E[.]/\text{var}(.)$. The parameters $\alpha\prime$ and $\beta\prime$ were obtained for each of the calculated values of $E[NO_3^- - N]$ leaching and $E[N_2O - N]$. Then, to introduce the uncertainty on $NO_3^- - N$ leaching and $N_2O - N$ emission estimations, the $NO_3^- - N$ leaching and $N_2O - N$ emission values were sampled from $gamma\left(\alpha'_{NO_3^-}, \beta'_{NO_3^-}\right)$ and $gamma\left(\alpha'_{N_2O}, \beta'_{N_2O}\right)$, respectively. Finally, in Eq. (19), $\Delta GHG_{\Delta N_2O} = GHG_{\Delta N_2O\,at\,E[AONR]} - GHG_{\Delta N_2O\,at\,selected\,case}$, while $\Delta N_{leaching} = N_{leaching\,at\,E[AONR]} - N_{leaching\,at\,selected\,case}$.

To estimate the benefit of N$_2$O–N emission reduction ($GHG_{\Delta N_2O}$), we normalized emissions into CO$_2$ equivalent (CO$_2$-e)[88]. Furthermore, for the $P_{GHG}$ in Eq. (19), it was assumed a price of 50 $/t CO$_2$-e based on Interagency Working Group's central estimate[89,90]. For computing $GHG_{\Delta fert}$, we based on the fact that the Haber–Bosch process is the most commonly utilized ammonia production method for N fertilization manufacturing[26,64]. The fact that the Haber–Bosch process emits 2.16 kg CO2 kg NH$_3^{-126}$ was also considered. Finally, for the $P_{leaching}$ in Eq. (19), it was assumed a price of 18.54 $/kg N[91].

For the selected case, the equivalent CO$_2$ reduction, the N leaching reduction, and the *SB* were obtained in terms of kg ha$^{-1}$ or $ ha$^{-1}$ for each scenario were scaled to evaluate potential benefits at the Corn-Belt level. The equivalent CO$_2$ reduction, the N leaching reduction, and the SB were calculated per state by multiplying their values and the area planted with maize for each of the considered states. We took the average maize planted area in the last five years (2019–2023) from the USDA-National Agricultural Statistics Service[3]. Finally, aggregated estimations for the US Corn Belt were obtained by assuming that the eight states studied followed simultaneously the same maize yield response to N fertilization. This means that maize planted in the area of the states behaved as each of the selected scenarios simultaneously. This assumption allowed us to provide potential cases of equivalent CO$_2$ reduction, N leaching reduction, and SB in the US Corn-Belt (Step 4 – Fig. 5).

### Reporting summary

Further information on research design is available in the Nature Portfolio Reporting Summary linked to this article.

## Data availability

All data used in this study are publicly available. The dataset of maize grain yield response to N used in this study are available in the ref. 92 database under accession code https://doi.org/10.5061/dryad.66t1g1k2g. The dataset of maize grain and fertilizer prices to define

probability distribution of the price ratio used in this study are available in the USDA-Economic Research Survey under the accession codes https://www.ers.usda.gov/data-products/fertilizer-use-and-price/documentation-and-data-sources and https://www.ers.usda.gov/data-products/season-average-price-forecasts. The literature review process about N-N$_2$O emissions and $N - NO_3^-$ leaching in the US Corn Belt data generated in this study are provided in the Supplementary Information file (Table S3). The data to build the plots presented in Fig. 3A and Figs. S1, S7–S9 can be reproduced from raw data and code that have already been shared in public repositories. The dataset for the area cultivated with maize presented in Fig. 1C can be obtained under the accession code https://croplandcros.scinet.usda.gov/. The plots shown in Figs. 1A, B and 5 are illustrations, no data was used to build those figures. Source data are provided with this paper in ref. 93 under accession code [https://doi.org/10.6084/m9.figshare.30524483]. Source data are provided with this paper.

## Code availability

The codes to fit the Bayesian quadratic and quadratic plateau models in this paper are available with open access at ref. 86 with accession code [https://doi.org/10.5281/zenodo.17868489].

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

## Acknowledgements

The authors thank Curtis Ransom for providing feedback on an early version of this manuscript. Science is a continuous process built by a community that shares knowledge. Therefore, the authors thank the efforts of those scientists who make their data freely available, contributing to faster scientific knowledge development.

## Author contributions

F.P. and I.A.C. conceived the study. F.P., I.A.C., E.A.D., P.V.V.P., J.R.S., and T.J.H. conceptualized the manuscript. F.P. analyzed the data and wrote the initial draft of the manuscript. F.P., E.A.D., K.G., A.J.E., H.E.B., P.V.V.P., T.J.H., J.R.S., and I.A.C. reviewed and edited the paper. E.A.D., K.G., A.J.E., and H.E.B. contributed disciplinary expertise to broaden the scope and interpretation of the results. F.P., E.A.D., K.G., A.J.E., H.E.B., P.V.V.P., T.J.H., J.R.S., and I.A.C. contributed to and approved the final version.

## Competing interests

The authors declare no competing interests.
