## [Peer Review file · Nature Communications]

Environmental and societal costs of maize production decrease by addressing the uncertainty in nitrogen rate recommendations

Corresponding Author: Dr Francisco Palmero

Version 1:

Reviewer comments:

Reviewer #1

(Remarks to the Author)

General Comments: This manuscript presents an interesting approach for analyzing existing nitrogen response curves to identify an appropriate nitrogen fertilizer application rate for maize production. The unique aspect of the approach is to consider the uncertainty in the rate recommendations based on the maximum return to nitrogen (MRTN) with implications for yield, as well as environmental and societal costs. While the manuscript is well conceived and would be of interest to the readership Nature Communication, there are few aspects of the approach that need to be addressed including the crop rotations that were evaluated as well as how the uncertainty could take into account response curves that do not fit the MRTN approach. Specific comments as well as a number of grammatical suggestions are as follows:

Line 53: and throughout the manuscript: be consistent with the abbreviation of nitrogen – use N.

Line 56: delete as (...to the soil in the form...

Line 63: Start a new paragraph with “Soil bacteria...”

Lines 128-129: revise: ... will be needed to deal with the way farmers traditionally think.

Lines 193-201: It is unclear if all these scenarios are continuous corn or some other rotation like corn-soybean. It would be useful to indicate what rotations are being considered.

Line 241 and throughout the manuscript: Are you expressing leaching load as nitrate or do you really mean nitrate-N. Since this manuscript is looking at N rates it would make more sense to express the amount of nitrate leached as nitrate-N. Same can be said about N₂O – are the emissions N₂O as N or as simply as N₂O gas?

Line 244: ... reduced NO₃- leaching by 0.83....

Line 271: ... was not considered.

Lines 279-281: See comments below for methods, but the data set used only considered N responses that fit the MRTN model i.e. with a plateau. Based on the data used, about 1 in 3 or one third of the sites showed a linear or non-plateau response and were disregarded. It is understandable why this was done for this study as the true N rate required is not known when a plateau is not attained or it could skew the MRTN to a much higher N rate. More discussion about this problem is warranted as this linear response is a likely reason why growers tend to over apply. While the analysis shows N rates can be reduced based on the uncertainty analysis, this is only true when a plateau response is achieved (about 66% of the time). As far as I could tell, the linear response data is not included in the uncertainty analysis.

Line 326: This may offset potential economic yield loss, but how does the farmer recoup this loss? There is some discussion below about this, but this statement still needs to be qualified, like through an incentive program.

Line 367: ...providing a significant yield decline....

Line 370-73: some rewording needed: In our study, we found that reducing the N application rate by 12% to 16% resulted in 7% to 13% reductions in N₂O emissions and 9% to 17% in NO₃-leaching with no or negligible risk (low probability and low expected value) of yield losses when accounting for the uncertainty of the N rate recommendations to farmers.

Line 400: do you mean the hectares of maize plated relative to hectares of soybeans has been altered? If so how, has it been altered higher or lower?

Lines 421-422: This is not entirely true. Many states give an "acceptable range"

Lines 447-449 and 459-461: How is previous crop taken into account? Does it matter for this analysis?

Line 483: was estimated...

Line 581: Define MCMC and add reference number.

Lines 597-602: See comments above. It is understandable why these sites were not included but these linear or non-plateau sites further add to the uncertainty of N rates. This could be addressed in the discussion section.

Lines 619-621: This is only the case when conditions are not conducive for N losses. There is a 1 in 3 chance that a plateau is not reached, hence a tendency to over-apply.

Line 650: The variances for...

Lines 663-665: revising needed: The fact that the Haber-Bosch process emits 2.16 kg CO₂ kg NH₃-1 26 was also considered.

(Remarks on code availability)

Reviewer #2

(Remarks to the Author)

(Remarks on code availability)

Reviewer #3

(Remarks to the Author)

The U.S. Corn Belt faces a severe problem of excessive nitrogen fertilizer application. This study quantifies the uncertainty embedded in current nitrogen rate recommendations to assess how much fertilizer can be reduced while keeping yield risk within an "acceptable yield risk" threshold. Results indicate that, by fully accounting for uncertainty, the Corn Belt could achieve a 12–16% reduction in N fertilizer application with negligible yield impacts, generating up to \$530 million in net social benefits. The work provides robust, probability-based evidence for designing "low-risk, high-benefit" nitrogen-management policies.

Overall, I find the paper innovative and policy-relevant. However, several revisions are necessary to meet publication standards:

1. The Abstract is structurally incomplete. Please add a concluding sentence that clearly summarizes the policy implications or the principal finding's significance for future research.

2. Reconsider figures selection, arrangement, and formatting, for instance:

Consider moving Fig. 1 to the Results section, insert sub-panel labels (a, b, c) and revise the title;

Given the extensive referencing of Supplementary Figures in the main Results section, consider merging key Supplementary Figures into the main text to enhance the clarity.

Line 267 references "FIG. 3B," yet Figure 3 lacks proper subfigure labels. Please standardize all figure and subfigure captions across the manuscript (e.g., Figure 2 currently has captions, while others do not).

3. Several expressions require clarification:

Lines 89-91: "Current fertilization programs, whether variable or non-variable rate, recommend the N rate that maximizes farmers' profit as a single (i.e., exact and unique value) N rate for the entire field or for a sub-region within a field."

Does the subsequent analysis only reference profit-maximizing scenarios? Does the study exclude the goal of production-maximizing recommendations? Clarify whether the two-phase approach (starting with AONR) implies that farmers in the study area have already adopted NONR?

Line 265: The rationale for selecting the two specific price points needs justification. Citations might be needed here.

Line 356-358: The reference to "Soil organic C" appears without sufficient preceding or following context. Please add

background or explanation to justify its inclusion at this point in the discussion.

Additionally, I would like to know what factors EONR considers. Does it incorporate environmental costs into farmers' production costs? Please explicitly discuss why the shift from AONR to EONR inherently yields environmental benefits (e.g., is it solely because it represents a reduction in applied N relative to the maximum possible yield?).

Some detailed suggestions as follows:

Line 151: This paragraph's formatting differs from others;

Line 154&156: "therefore" is used twice consecutively;

Line 330: The abbreviation "SB" is used without prior definition;

Line 380: Consider adding references;

Line 403: it should be BMPs;

Line 626: Does the formula require multiplication symbol?

(Remarks on code availability)

Reviewer #4

(Remarks to the Author)

A major source of nitrogen leaching into surface and ground waters is, in general, agriculture, and maize production is a primary contributor within agriculture. Extensive studies have been conducted to develop nitrogen rate recommendations for maize. These recommendations have been made historically for a state or a region within a state. The advent of precision agriculture has enabled recommendations to be implemented at finer spatial scales. However, variability in maize fields, weather, management, and other factors make it impossible to publish precise regional or state nitrogen recommendations. The authors consider the uncertainty associated with nitrogen rate recommendations. They suggest an approach to reducing the use of nitrogen with a quantifiable amount of risk in terms of yield loss for the producer and the associated societal benefit. This review focuses on the statistical aspects of the paper.

The statistical analysis is creative, strong, and well presented. The following comments focus on helping the reader understand the results.

It would be helpful to present, or at least point to, the form of the quadratic plateau model prior to equation (1). Otherwise, readers, including this one, may spend time wondering why there is no negative sign present in (1), only to learn on the following page that the quadratic term in the model has a negative sign.

As measures are introduced in the results section, illustrations could improve the readability. Three specific examples are provided, and the authors should consider others:

1. In the paragraph beginning on line 138, basic measures that are then used throughout the paper are provided. It would be helpful to present, for one field, the estimates of $E[AONR]$ and $E[EONR]$ followed by the computations of the uncertainties, both in terms of N rates and proportions.
2. In the paragraph beginning on line 151, the processes behind estimating the probability of losing a percentage of grain yield need to be better explained, again preferably with an illustration, especially since the related figure is in the supplementary materials.
3. Beginning on line 170, the case 2 reduction in grain yield was "an expected grain yield of 16,708 kg/ha – 8,438 kg/ha at the expected value of the EONR." Again, helping the reader understand each of the numbers would be nice. This reader assumes that 16,708 kg/ha is the EONR and 8,438 kg/ha is the reduction. If that is not correct, perhaps it will help the authors understand where improvements in clarity can be made. If it is correct, then the authors should realize that this is not evident from the presentation.

This is excellent work, and the authors are to be commended. Others will benefit more from reading the paper if the authors help them by providing illustrations (using only a sentence or two in each case) and a few clarifying words. Thank you for the opportunity to read the paper.

(Remarks on code availability)

Version 2:

Reviewer comments:

Reviewer #1

(Remarks to the Author)

I read through the rebuttal and the revised manuscript and the authors addressed my comments thoughtfully and thoroughly. They should be commended for a job well done. In my opinion, the revised manuscript now is acceptable for publication.

(Remarks on code availability)

Reviewer #2

(Remarks to the Author)

Co-reviewed

(Remarks on code availability)

Reviewer #3

(Remarks to the Author)

The authors have done an excellent job of addressing our previous questions. After reviewing the refined manuscript, we have some remaining minor comments.

Line 30: Does "their" refer to the "Additional N reduction"?

The statement that the practice leads to a "high" risk of yield loss seems to contradict the expression that the proposed reduction is possible "with negligible risk of maize yield losses." The authors should clarify the source and extent of this "high" risk.

The term "unsustainable" might be less precise than "unacceptable" in describing the induced farmer behavior.

Please consider rewriting the sentence, perhaps as: "the potential risk of yield loss makes this practice unacceptable for farmers".

Line 260: The subtitle formatting for Figure 3 appears to be different from the others.

(Remarks on code availability)

Reviewer #4

(Remarks to the Author)

The authors have carefully considered each comment on the initial review of the paper. This revision is more readable and provides increased clarity to the methods as well as the strengths and weaknesses of the work.

Thank you for the opportunity to read your work.

(Remarks on code availability)

Response to reviewers' comments

Reviewer #1 (Remarks to the Author):

General Comments: This manuscript presents an interesting approach for analyzing existing nitrogen response curves to identify an appropriate nitrogen fertilizer application rate for maize production. The unique aspect of the approach is to consider the uncertainty in the rate recommendations based on the maximum return to nitrogen (MRTN) with implications for yield, as well as environmental and societal costs. While the manuscript is well conceived and would be of interest to the readership Nature Communication, there are few aspects of the approach that need to be addressed including the crop rotations that were evaluated as well as how the uncertainty could take into account response curves that do not fit the MRTN approach. Specific comments as well as a number of grammatical suggestions are as follows:

1) Line 53: and throughout the manuscript: be consistent with the abbreviation of nitrogen – use N.

Response: Thank you for noticing this, and for providing this suggestion. We have changed the word “nitrogen” by “N” in all the document, after defining the meaning of N in line 34. We kept the word “nitrogen” in titles and in figure captions (to make figures self-explanatory).

2) Line 56: delete as (...to the soil in the form...

Response (line 59): Thank you. We deleted “...to the soil as in the form ...”. We made further changes to improve the grammar of the sentence after erasing that fragment.

3) Line 63: Start a new paragraph with “Soil bacteria...”

Response (line 67): Thank you for this suggestion. This comment was very helpful in improving the reading of the manuscript. We have split the original paragraph starting a new one with “Soil bacteria...”.

4) Lines 128-129: revise: ... will be needed to deal with the way farmers traditionally think.

Response (lines 138 and 139): Thank you for your recommendation. We have changed the original sentence. The new version is much clearer, thank you.

6) Lines 193-201: It is unclear if all these scenarios are continuous corn or some other rotation like corn-soybean. It would be useful to indicate what rotations are being considered.

Response (lines 251 and 252): Thank you for noticing this. It will be helpful for readers to have this information when it comes to interpreting the results. There was just one site in entire data base with continuous soybean in the four seasons previous to the analyzed maize experiment. This was Scenario A. Scenarios B and C had corn-soybean rotations. We have added this information in the main text.

For a more complete background, among the 49 site-year combinations in this open database used in our study, there was one site (trial 10, in Troth, MO, year 2014) with continuous soybean in the four years previous to the corn experiment analyzed in our study. Furthermore, there was one site (trial 12, in Durbin, ND, year 2014) with continuous corn in the four years previous to the corn experiment analyzed in our study. The rest of the site-year combinations in the database correspond to corn-soybean rotations in the four previous seasons. This means 96% of the analyzed experiments corresponded to corn-soybean rotations.

7) Line 241 and throughout the manuscript: Are you expressing leaching load as nitrate or do you really mean nitrate-N. Since this manuscript is looking at N rates it would make more sense to express the amount of nitrate leached as nitrate-N. Same can be said about N₂O – are the emissions N₂O as N or as simply as N₂O gas?

Response: Thank you for noticing this detail. The equations we have used to estimate nitrogen losses through leaching and nitrous oxide emissions considered nitrogen balance as the predictor variable. The scale of the obtained predictions (response variable), for both leaching and gas emission, was in kilograms of nitrogen per hectare (kg N ha⁻¹). Therefore, in all the manuscript we express leaching load as nitrate-N ($NO_3^- - N$) and nitrous oxide emissions as $N_2O - N$. We modified this in the entire article. We replaced NO_3^- by $NO_3^- - N$ to make it clear that we are referring to nitrogen losses from nitrate when dealing with leaching. We have also replaced N_2O

by N_2O -N to clarify that we refer to nitrogen losses from nitrous oxide when dealing with nitrogen oxide gas emissions.

8) Line 244: ... reduced NO_3 - leaching by 0.83....

Response (line 305): Thank you. We have replaced “in” with “by”.

9) Line 271: ... was not considered.

Response (line 333): Thank you for highlighting this grammar error. We changed “were” to “was”.

10) Lines 279-281: See comments below for methods, but the data set used only considered N responses that fit the MRTN model i.e. with a plateau. Based on the data used, about 1 in 3 or one third of the sites showed a linear or non-plateau response and were disregarded. It is understandable why this was done for this study as the true N rate required is not known when a plateau is not attained or it could skew the MRTN to a much higher N rate. More discussion about this problem is warranted as this linear response is a likely reason why growers tend to over apply. While the analysis shows N rates can be reduced based on the uncertainty analysis, this is only true when a plateau response is achieved (about 66% of the time). As far as I could tell, the linear response data is not included in the uncertainty analysis.

Response (lines 359-366 and 700-708): Thank you for bringing this very interesting point. You are correct; the flat and linear responses sites were not included in the uncertainty analysis. We have acknowledged this by adding two sentences into the manuscript.

To complete our response, we did not consider the sites when: (i) the yield plateau was achieved at a N rate ($E[\text{AONR}]$) greater than the highest rate implemented in the studies and (ii) the sites showed Gelman-Rubin diagnostic greater than 1.02 for the AONR and/or EONR.

If we had considered the sites where the yield plateau was achieved at a N rate ($E[\text{AONR}]$) greater than the highest rate implemented in the studies, then we would have been making an extrapolation to find the optimal N rates. This would have led to larger uncertainty estimation on the predicted value of the optimal N rates, and consequently, a larger N reduction would have

been possible with small risk of yield losses. However, this scenario would not represent a case in real life, since our inability to find the optimal rates precisely would not be highly related to the uncertainty attached to weather, soil, and crop demand variability (which is the real uncertainty), but it would be attached to a poor experimental design (lack of exploring more N rates) or data quality. Therefore, we opted for a more conservative approach by removing those sites.

On the other hand, as the reviewer mentioned, the sites where the Gelman-Rubin diagnostic was greater than 1.02 for the AONR and/or EONR had either no response (flat horizontal line) or a linear response. To give a better perspective, 11 sites (out of 49 studied, i.e., ~22% of the sites) showed a flat response (North Dakota: T12, T27, T28, T44, T45; Nebraska: T30, T47; Wisconsin: T31, T32, T48, T49). On the other hand, 6 sites (out of 49 studied, i.e., ~12% of the sites) showed a linear response (Illinois: T3; Minnesota: T39; Missouri: T25, T26; Nebraska: T13, T29).

In both cases (flat and linear responses), the optimal nitrogen rates could not be determined. This meant that although we were able to compute an expected value and its associated uncertainty, we could not trust those values because the model did not converge for the computation of the optimal N rates. Therefore, in those cases, we would have had considerably high under- or over-estimations of the expected values of the optimal N rates with high uncertainty, leading (once again) to big, unreal N reduction with small risk of yield losses. However, as the reviewer mentioned, these two alternatives do not have the same effect on farmers' decision making process. When there is a flat response, farmers, extensionists, and crop advisors recognize that there is no need to apply N. However, in linear responses farmers, extensionists, and crop advisors realized that more N would lead to higher yields, but the appropriate optimal N rate cannot be precise and accurately estimated. Therefore, linear responses (~12% of the sites) would increase the uncertainty of N rates leading to overapplications by farmers.

As a summary, we have considered these two approaches for not including the sites having these characteristics to make a more conservative and realistic analysis of potential N fertilizer rate reductions and their associated risks of yield loss (probability of losing yield and its expected yield loss).

We have not explained this clearly and connected the linear and flat responses as reasons behind potential increases in the uncertainty of the N rate recommendations leading farmers to overapply N fertilizers. Now, in the Methods section under the subtitle "Assessing N reduction feasibility" we have mentioned the reasons explaining why we have removed the sites that were not included in the final analyses of potential N rates reductions and their associated risk of yield loss (lines 700-

708). Furthermore, in the discussion section (under the subtitle “Uncertainty on the fertilizer N rate recommendations”), we have suggested the linear responses as potential reasons for increasing the uncertainty of N rates leading to overapplications by farmers (lines 359-366).

11) Line 326: This may offset potential economic yield loss, but how does the farmer recoup this loss? There is some discussion below about this, but this statement still needs to be qualified, like through an incentive program.

Response (lines 408-410): Thank you for this suggestion. We completely agree with reviewer’s comment; a statement was needed to better present the idea in that paragraph (lines 410-412).

12) Line 367: ...providing a significant yield decline....

Response (line 455): Thank you. We have made that correction.

13) Line 370-73: some rewording needed: In our study, we found that reducing the N application rate by 12% to 16% resulted in 7% to 13% reductions in N₂O emissions and 9% to 17% in N₀₃- leaching with no or negligible risk (low probability and low expected value) of yield losses when accounting for the uncertainty of the N rate recommendations to farmers.

Response (lines 459-462): Thank you for pointing out these changes for clarity. We have applied the suggested rewriting.

14) Line 400: do you mean the hectares of maize planted relative to hectares of soybeans has been altered? If so how, has it been altered higher or lower?

Response (lines 489-490): Thank you for this comment; that statement needs to be clarified. As the reviewer mentioned, we referred to modifications in the hectares of maize planted relative to the hectares of soybeans. This maize:soybean ratio has decreased, i.e., more farmers decided to plant soybean due to a considerable increment in N fertilizer prices. We have clarified that in lines 491-492.

15) Lines 421-422: This is not entirely true. Many states give an “acceptable range”

Response (lines 511-513): Thank you for noticing this. The reviewer is completely right; many states following the MRTN approach provide farmers with a range of optimal N rates. However, the limits of the range are arbitrarily defined by considering 1 \$/acre below and above the MRTN value. Therefore, the MRTN is subjectively defining a “most profitable range” within 1 \$/acre MRTN. In our study, we suggest the use of true (objective) statistical ranges of most probable values for optimal N rates based on the probability distribution of these quantities. This approach enabled us to analyze different cases of N rate reductions and their potential yield losses in a probabilistically way. We have modified the statement to consider the MRTN approach where a range of optimal N rates is subjectively provided based on profits.

16) Lines 447-449 and 459-461: How is previous crop taken into account? Does it matter for this analysis?

Response: Thank you for pointing this out. It is well known that the previous crop can influence the optimum N rate required in a region. However, we hypothesize that the difference in optimal N rates between corn and soybean as previous crops is small compared to the overall uncertainty in the estimates, making the effect of crop rotation less relevant compared to the level of uncertainty to define the optimal N rate for corn. This idea can be roughly seen (though not fully statistically accurate) in the box sizes shown in Baum et al. (2025). Therefore, with all due respect, we believe that considering the crop rotation is not particularly critical for the specific analysis in our study. Still, we have added a note about this as a direction for future research in the discussion section (line 521-523). We have also clarified in the methods section that 96% of the studied sites are all derived from the same type of crop rotation, corn-soybean rotations (lines 551-553), and thus, the potential effect of crop rotation is not a relevant aspect for our analyses.

17) Line 483: was estimated...

Response (line 580): Thank you. We replaced “were” by “was”.

18) Line 581: Define MCMC and add reference number.

Response (line 678): Thank you. We have addressed this comment.

19) Lines 597-602: See comments above. It is understandable why these sites were not included but these linear or non-plateau sites further add to the uncertainty of N rates. This could be addressed in the discussion section.

Response: Thank you for these comments. We have addressed them and provided a response in comment number 10. We appreciate the reviewer bringing this topic, since now an extra source of uncertainty potentially explaining N fertilizer overapplications was added in different sections of the manuscript.

20) Lines 619-621: This is only the case when conditions are not conducive for N losses. There is a 1 in 3 chance that a plateau is not reached, hence a tendency to over-apply.

Response: Thank you for your comment. The reviewer is completely right. When the grain yield response to N is accurately described by a model with a decreasing slope (e.g., quadratic-plateau functions), reduction on the N rate lead to an increase in the secant slope between yield at zero N application and yield at a given N rate, i.e., the agronomic use efficiency of the fertilizer. Therefore, under these conditions, reducing N rates inherently improves fertilizer use efficiency. In contrast, when the yield response to N is best represented by a model with a constant slope (e.g., linear response, which represented 12% of the analyzed sites in our study), reduction on the N rate does not lead to changes in the secant slope between yield at zero N application and yield at a given N rate. In such cases, reducing N does not increase fertilizer use efficiency and may considerably increase the risk of yield loss. Consequently, these conditions introduce greater uncertainty, which might lead to N overapplications. We illustrated this in the plot below.

$$\text{Fertilizer use efficiency} = \frac{\text{Yield at Nrate} - \text{yield at 0N}}{\text{N rate}}$$

We have added a sentence in lines 727-728, to clarify that lower N rates mean higher fertilizer use efficiency is not universally true, but when the slope of the fitted model is decreasing. Furthermore, we have added a paragraph to the discussion section based on this comment (lines 367-376).

21) Line 650: The variances for...

Response (line 757): Thank you. We have added the letter “s”.

22) Lines 663-665: revising needed: The fact that the Haber-Bosch process emits 2.16 kg CO₂ kg NH₃-1 26 was also considered.

Response (lines 771-772): Thank you. We have moved “was also considered” from the beginning to the end of the sentence.

Reviewer #2

Co-reviewed

Reviewer #3 (Remarks to the Author):

The U.S. Corn Belt faces a severe problem of excessive nitrogen fertilizer application. This study quantifies the uncertainty embedded in current nitrogen rate recommendations to assess how much fertilizer can be reduced while keeping yield risk within an “acceptable yield risk” threshold. Results indicate that, by fully accounting for uncertainty, the Corn Belt could achieve a 12–16% reduction in N fertilizer application with negligible yield impacts, generating up to \$530 million in net social benefits. The work provides robust, probability-based evidence for designing "low-risk, high-benefit" nitrogen-management policies.

Overall, I find the paper innovative and policy-relevant. However, several revisions are necessary to meet publication standards:

1) The Abstract is structurally incomplete. Please add a concluding sentence that clearly summarizes the policy implications or the principal finding's significance for future research.

Response (lines 29-32): Thank you. We have added a sentence summarizing the policy implications of our results and conclusions.

2) Reconsider figures selection, arrangement, and formatting, for instance:

2.1) Consider moving Fig. 1 to the Results section, insert sub-panel labels (a, b, c) and revise the title;

Response (lines 110-117): Thank you for this comment. We have added sub-panel labels to help readers focus on the key points mentioned in the main text, and we have changed the figure title. We have also added references to each panel in the main text. Regarding moving the figure to the Results section, with all due respect, the figure is a schematic representation. In other words, it does not present any results from our study, but helps to frame the main issue that the paper is bringing forward. We believe that placing it in the Results section could be confusing for readers, since the figure does not actually contain any study results.

2.2) Given the extensive referencing of Supplementary Figures in the main Results section, consider merging key Supplementary Figures into the main text to enhance the clarity.

Response (lines 228-239): Thank you, we have moved supplementary figure 6 to the main text to enhance clarity. This is a figure without sub-panels. Therefore, no letters were added for reference. Although we would like to move more relevant figures from the supplementary materials to the main text, we believe this is not possible due to the size of the figures and due to the fact that if Fig. S2 is moved then Fig. S3 should be, and if Fig. S4 is moved then Fig. S5 should be, because each figure in those pairs are equally important. On the other hand, the supplementary figures 2, 3, 4, and 5 depict the results for all sites analyzed in the study. Three sites were selected as representative of the main different scenarios encountered for the entire dataset (called Scenarios A, B, and C), and the main findings are based on the representative sites. Otherwise, it would be very difficult and unclear to transmit the results and conclusions of our study.

On the other hand, based on reviewer's number 4 comments, to ensure clarity and facilitate readers' interpretation of the results, we have added a paragraph illustrating the calculations of the results for one particular field study. These descriptions were added in the subsections that rely the most on supplementary materials, which are "Fertilizer N rates: point and uncertainty estimations" and "Feasibility of Nitrogen fertilization reductions" within the Result section.

2.3) Line 267 references "FIG. 3B," yet Figure 3 lacks proper subfigure labels. Please standardize all figure and subfigure captions across the manuscript (e.g., Figure 2 currently has captions, while others do not).

Response: Thank you for this comment. We have made these changes in the figures and in the main text when corresponding.

3) Several expressions require clarification:

3.1) Lines 89-91: "Current fertilization programs, whether variable or non-variable rate, recommend the N rate that maximizes farmers' profit as a single (i.e., exact and unique value) N rate for the entire field or for a sub-region within a field."

Response (lines 94-97): Thank you for your suggestion. We have rewritten the sentence to make it clearer.

3.2) Does the subsequent analysis only reference profit-maximizing scenarios? Does the study exclude the goal of production-maximizing recommendations? Clarify whether the two-phase approach (starting with AONR) implies that farmers in the study area have already adopted NONR?

Response: Thank you for this comment; it helped to clarify some points. In lines 85-87 we mentioned that the AONR and EONR are the nitrogen rates maximizing grain yield and farmers' profit, respectively. Then, in the paragraph starting in line 126, our analysis proposes, in Phase I, N reductions going from the expected value of the AONR to the expected value of the EONR. In Phase II, the suggested reductions go from the expected value of the EONR to some quantile of the probability distribution of the EONR (studied by analyzing four cases of N reduction). Therefore, our subsequent analysis does not refer only to profit-maximizing scenarios, but it starts with grain yield maximization and ends in some value of optimal N rate within the probability distribution of the rates maximizing farmers' profit (EONR). We have added a new sentence in lines 134-136 to clarify that our analysis encompasses both, grain yield and profits maximizations.

Regarding whether the two-phase approach (starting with AONR) implies that farmers in the study area have already adopted AONR; in lines 129-131, we have mentioned that a high proportion of farmers apply more N than the recommended N rate, and the actual applied N rate is usually above the $E[AONR]$. This is supported by previous studies mentioned below in this response. Then, as reviewer said, in our analysis we assume (or imply) that farmers have already adopted AONR. Although we are missing the farmers that apply more than the AONR (which is a high proportion), assuming they apply AONR is the middle ground in between those applying the recommended rates (i.e., EONR) and those applying more than AONR. Therefore, assuming farmers apply the expected value of the AONR allowed us to be realistic and conservative in our analysis. We have added an extra sentence in lines 139-140 hoping to make this implication/assumption clearer.

References:

Yadav, S. N., Peterson, W. & Easter, K. W. Do farmers overuse nitrogen fertilizer to the detriment of the environment? *Environ Resource Econ* 9, 323–340 (1997).

Houser, M. Farmer Motivations for Excess Nitrogen Use in the U.S. Corn Belt. *Case Studies in the Environment* 6, 1688823 (2022).

Sellars, S. C., Schnitkey, G. D. & Gentry, L. F. Do Illinois Farmers Follow University-Based Nitrogen Recommendations? (2020)

3.3) Line 265: The rationale for selecting the two specific price points needs justification. Citations might be needed here.

Response: Thank you. We agree with the reviewer about the fact that citations might be needed here. However, line 265 was in the Result section, unless results are blended with discussion, scientific writing theory does not recommend adding citations in the Result section. This does not mean that the use of these two specific price points does not need to be justified, but the justification and citations are in the Methods section. Since these prices come from other studies and we use them to calculate new variables in our study, readers should refer to the Methods section. We cited the studies that justify the use of these prices in the Methods section in lines 767 and 773. We have added clarification in between parentheses to refer readers to the Methods section for more information (line 327).

3.5) Line 356-358: The reference to "Soil organic C" appears without sufficient preceding or following context. Please add background or explanation to justify its inclusion at this point in the discussion.

Response (lines 439-444): Thank you for catching this. We have added information to clarify that soil organic carbon is connected to capturing carbon from the atmosphere, helping to reduce global warming. This is to highlight that we are proposing changes in N fertilization management to reduce greenhouse gas emissions without affecting other agriculture-related alternatives to reduce greenhouse gases concentrations in the atmosphere. We have added background and explanation to justify the inclusion of soil organic carbon at that point of the discussion (lines 439-444).

3.6) Additionally, I would like to know what factors EONR considers. Does it incorporate environmental costs into farmers' production costs? Please explicitly discuss why the shift from AONR to EONR inherently yields environmental benefits (e.g., is it solely because it represents a reduction in applied N relative to the maximum possible yield?).

Response: Thank you. Regarding the factors EONR considers, we have explained this in the Methods section, under the subtitle “Statistical analysis” (lines 602-616). There, it can be seen that the EONR does not consider environmental costs into farmers’ production costs.

On the other hand, we have clarified in lines 446-448 within the discussion section that, as you said, the shift from the E[AONR] to E[EONR] inherently yields environmental benefits just because it represents a reduction in the N rate relative to that maximizing grain yield.

4) Some detailed suggestions as follows:

4.1) Line 151: This paragraph’s formatting differs from others;

Response (line 191): Thank you. We have corrected the formatting.

4.2) Line 154&156: “therefore” is used twice consecutively;

Response (line 196): Thank you. We have changed the second one by a synonym.

4.3) Line 330: The abbreviation “SB” is used without prior definition;

Response (line 413): Thank you. We have made corresponding corrections.

4.4) Line 380: Consider adding references;

Response (line 469): Thank you. We have added references.

4.5) Line 403: it should be BMPs;

Response (line 493): Thank you. We have added the letter “s”.

4.6) Line 626: Does the formula require multiplication symbol?

Response: Thank you for catching this. In mathematics, there is a method known as “implied multiplication”. According to this method, we can express multiplications without a symbol by placing one number or variable next to parentheses, or by writing variables and numbers next to each other. According to this method, for the equation the reviewer mentioned (equation number 19), there is no need to place a multiplication symbol since we are using general variable notation and parentheses. We have applied the implied multiplication method consistently in all the equations in our manuscript. For example, in equations 20 and 21, no multiplication symbol is needed because we wrote numbers and variables next to each other.

Reviewer #4 (Remarks to the Author):

A major source of nitrogen leaching into surface and ground waters is, in general, agriculture, and maize production is a primary contributor within agriculture. Extensive studies have been conducted to develop nitrogen rate recommendations for maize. These recommendations have been made historically for a state or a region within a state. The advent of precision agriculture has enabled recommendations to be implemented at finer spatial scales. However, variability in maize fields, weather, management, and other factors make it impossible to publish precise regional or state nitrogen recommendations. The authors consider the uncertainty associated with nitrogen rate recommendations. They suggest an approach to reducing the use of nitrogen with a quantifiable amount of risk in terms of yield loss for the producer and the associated societal benefit. This review focuses on the statistical aspects of the paper.

The statistical analysis is creative, strong, and well presented. The following comments focus on helping the reader understand the results.

1) It would be helpful to present, or at least point to, the form of the quadratic plateau model prior to equation (1). Otherwise, readers, including this one, may spend time wondering why there is no negative sign present in (1), only to learn on the following page that the quadratic term in the model has a negative sign.

Response: Thank you for catching this detail. The reviewer is completely right, as the quadratic (or quadratic plateau) model is usually written, the first derivative of the model with respect to x is $\beta_1 + 2\beta_2x$. When using the derivative to find the value of x where the function achieves its maximum (or minimum), we end up with $x = \frac{\beta_1}{-2\beta_2}$. However, considering previous knowledge on this area of study, we wrote the model such that we obtain a flat line or a concave downward line. This means that the β_2 parameter can be very close to zero or negative. Therefore, by adding this restriction to the model, the maximum is achieved at $x = \frac{\beta_1}{2\beta_2}$ instead of a maximum or a minimum at $x = \frac{\beta_1}{-2\beta_2}$. However, as the reviewer mentioned, the restriction imposed over the model is presented below the AONR and EONR equations.

To clarify this, we have referred to the shape of the quadratic plateau model we are considering in the Introduction section (i.e., before equation 1), this is in lines 92-93. In the Methods section, we have added reference to the Step 1 in Fig 5, where the shape of the quadratic plateau model can be seen (line 597). We have also added the minus symbol in the calculations of the AONR and EONR, since these would be the result when expressing the model without restrictions (lines 601 and 605). Then, when readers see the restriction imposed in the model will automatically know that β_2 is negative, so the final version of equations 1 and 2 would be $AONR = \frac{\beta_1}{2\beta_2}$ and $EONR = \frac{\beta_1 - PR}{2\beta_2}$.

2) As measures are introduced in the results section, illustrations could improve the readability. Three specific examples are provided, and the authors should consider others:

2.1) In the paragraph beginning on line 138, basic measures that are then used throughout the paper are provided. It would be helpful to present, for one field, the estimates of E[AONR] and E[EONR] followed by the computations of the uncertainties, both in terms of N rates and proportions.

Response (lines 149-159): Thank you for this comment. We appreciate it because the changes will facilitate readers' interpretation of the results. We have added a paragraph illustrating the calculations for one particular field study.

2.2) In the paragraph beginning on line 151, the processes behind estimating the probability of losing a percentage of grain yield need to be better explained, again preferably with an illustration, especially since the related figure is in the supplementary materials.

Response (lines 173-190): Thank you for this suggestion. We believe the interpretation of the results should improve after adding this illustration. We really appreciate reviewer's comment.

2.3) Beginning on line 170, the case 2 reduction in grain yield was “an expected grain yield of 16,708 kg/ha – 8,438 kg/ha at the expected value of the EONR.” Again, helping the reader understand each of the numbers would be nice. This reader assumes that 16,708 kg/ha is the EONR and 8,438 kg/ha is the reduction. If that is not correct, perhaps it will help the authors understand where improvements in clarity can be made. If it is correct, then the authors should realize that this is not evident from the presentation.

Response (lines 209-214): Thank you once again for helping us to facilitate the interpretation of the results of this study. We have clarified this by rewriting the fraction of the text the reviewer highlighted. The changes can be seen in lines 209-214.

We did consider other sections of the results where illustrations or rewriting could improve the readability. In connection with reviewer 2, for improving the interpretation and readability of the “Feasibility of Nitrogen fertilization reductions” sections within the Results, we have moved Fig S6 to the main text. Furthermore, we have carefully read the results section looking for more sections where illustrations or rewriting could improve readability. However, based on reviewer’s 3 suggestions, we found a cascade effect in the sense that the changes made for illustrating calculations automatically facilitated the interpretation of the rest of the results. Therefore, we greatly appreciate the recommendations of illustrating calculations and pick one site for showing them.

This is excellent work, and the authors are to be commended. Others will benefit more from reading the paper if the authors help them by providing illustrations (using only a sentence or two in each case) and a few clarifying words. Thank you for the opportunity to read the paper.

Response: Thank you. We honestly appreciate reviewer’s comments; they helped to facilitate the interpretation of the results in our study and to provide a better understanding of the statistics implemented.

Thanks to all the comments and suggestions, clearly, they helped to improve the quality of this work. Looking forward to a positive outcome.

Best regards,

Authors.

Response to reviewers' comments

Reviewer #1 (Remarks to the Author):

I read through the rebuttal and the revised manuscript and the authors addressed my comments thoughtfully and thoroughly. They should be commended for a job well done. In my opinion, the revised manuscript now is acceptable for publication.

Response: Thank you. We appreciate reviewer's time devoted to reviewing our research article. Their comments have been really helpful to improve the quality of the manuscript.

Reviewer #2 (Remarks to the Author):

Co-reviewed

Reviewer #3 (Remarks to the Author):

The authors have done an excellent job of addressing our previous questions. After reviewing the refined manuscript, we have some remaining minor comments.

Response: Thank you. We appreciate the reviewer's time and helpful comments, which have improved our manuscript.

Line 30: Does "their" refer to the "Additional N reduction"?

Response (lines 30-31): Thank you for noticing. Yes, "their" refers to the "additional N reductions". We have clarified this by replacing "their" by "the high risk of yield loss associated with additional N reductions makes this practice unsustainable for farmers".

The statement that the practice leads to a "high" risk of yield loss seems to contradict the expression that the proposed reduction is possible "with negligible risk of maize yield losses." The authors should clarify the source and extent of this "high" risk.

Response (lines 30-31): Thank you. The reviewer is right. However, with the modification we made based on the previous comment, now it is clear that the high risk is associated with additional N reductions beyond the ones proposed in this study. That is, the ones proposed in this study have low risk, but the additional ones have high risk.

The term "unsustainable" might be less precise than "unacceptable" in describing the induced farmer behavior.

Please consider rewriting the sentence, perhaps as: "the potential risk of yield loss makes this practice unacceptable for farmers".

Response (line 32): Thank you. We have replaced the word "unsustainable" by "unacceptable".

Line 260: The subtitle formatting for Figure 3 appears to be different from the others.

Response: Thank you. We have checked this. The formatting was the same as the others.

Reviewer #4 (Remarks to the Author):

The authors have carefully considered each comment on the initial review of the paper. This revision is more readable and provides increased clarity to the methods as well as the strengths and weaknesses of the work.

Thank you for the opportunity to read your work.

Response: Thank you. Reviewer's comments have been really helpful to improve the quality of the manuscript. We appreciate reviewer's time devoted to reviewing our research article.